# Sounding that Object: Interactive Object-Aware Image to Audio Generation

**Tingle Li** [1]  **Baihe Huang** [1]  **Xiaobin Zhuang** [2]  **Dongya Jia** [2]  **Jiawei Chen** [2]  **Yuping Wang** [2]  **Zhuo Chen** [2]
**Gopala Anumanchipalli** [1]  **Yuxuan Wang** [2]

## Abstract

Generating accurate sounds for complex audio-visual scenes is challenging, especially in the presence of multiple objects and sound sources. In this paper, we propose an *interactive object-aware audio generation* model that grounds sound generation in user-selected visual objects within images. Our method integrates object-centric learning into a conditional latent diffusion model, which learns to associate image regions with their corresponding sounds through multi-modal attention. At test time, our model employs image segmentation to allow users to interactively generate sounds at the *object* level. We theoretically validate that our attention mechanism functionally approximates test-time segmentation masks, ensuring the generated audio aligns with selected objects. Quantitative and qualitative evaluations show that our model outperforms baselines, achieving better alignment between objects and their associated sounds. Project site: `https://tinglok.netlify.app/files/avobject/`.

## 1. Introduction

Humans naturally perceive the world as ensembles of distinct objects and their associated sounds (Bregman, 1994). For example, in a busy city street (Figure 1), we can identify sounds from multiple objects, such as honks, footsteps, and chatters. However, replicating such object-level specificity remains challenging for computational models. Despite notable advances in audio generation (Van Den Doel et al., 2001; Kong et al., 2019; Yang et al., 2023), existing methods often generate holistic soundscapes that fail to accurately reproduce the distinct sounds of specific objects (Pijanowski et al., 2011). In complex scenes, models may either *forget* subtle sounds (e.g., footsteps) or *bind* co-occurring events

(e.g., crowd noise and wind) even when only one is intended, leading to inaccurate sound textures (McDermott & Simoncelli, 2011).

Recent progress in vision-based models (Sheffer & Adi, 2023) relies on analyzing the entire visual scene to produce a single soundtrack, but this broad perspective may overlook subtle yet important sound sources. Text-based models (Liu et al., 2023), on the other hand, face difficulties when a prompt represents multiple events, either omitting certain sounds or conflating them with others due to entangled feature correlations (Wu et al., 2023). While manually reweighting individual sound events in the diffusion latent (Xue et al., 2024) can mitigate these issues, it remains labor-intensive and impractical for large-scale applications. Fundamentally, these challenges arise because real-world sounds are often *imbalanced* and *confounding* in complex scenes, making it difficult to disentangle distinct sound sources.

To overcome these limitations, we propose an *interactive object-aware audio generation* model that grounds each generated sound in a specific visual object. Inspired by how humans parse complex soundscapes (Gaver, 1993), our model not only processes the overall scene context (e.g., a city street) but also decouples separate events (e.g., honks, footsteps). Drawing on object-centric learning (Greff et al., 2019), we build our model upon a conditional audio generation framework (Liu et al., 2023) and introduce multi-modal dot-product attention (Vaswani et al., 2017) to learn sound-object associations through self-supervision (Zhao et al., 2018; Afouras et al., 2020), which fundamentally overcomes the problem of *forgetting* or *binding* sound events.

To provide finer control and interactivity, we leverage segmentation masks (Kirillov et al., 2023) to convert user queries into attention maps at test time, allowing users to select specific objects in an image (e.g., car shapes) to generate the corresponding sounds (e.g., engine sounds) with simple mouse clicks. Since these masks guide the model to focus on objects of interest, even subtle sound events can be captured more accurately than with scene-wide analysis alone. Moreover, because the entire image still informs the generation process, selecting multiple objects naturally blends their sounds into a consistent environment, rather

---

[1]University of California, Berkeley [2]ByteDance Inc. Correspondence to: Tingle Li <tingle@eecs.berkeley.edu>.

*Proceedings of the 42nd International Conference on Machine Learning*, Vancouver, Canada. PMLR 267, 2025. Copyright 2025 by the author(s).

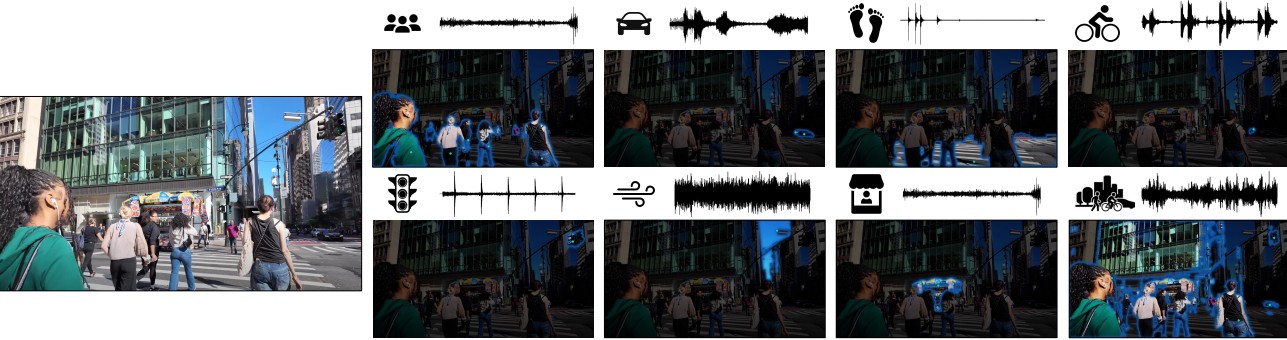

Figure 1: **Interactive object-aware audio generation**. We generate sound aligned with specific visual objects in complex scenes. Users can select one or more objects in the scene using segmentation masks, and our model generates audio corresponding to the selected objects. Here, we show a busy street with multiple sound sources (left). After training, our model generates object-specific audio (right), such as crowd noise for people, engine sounds for cars, and blended audio for multiple objects.

than merely layering independent audio clips.

Through quantitative evaluations and human perceptual studies, we show that our model generates more complete and contextually relevant soundscapes than existing baselines. Additionally, we provide qualitative results and theoretical analysis to demonstrate that our object-grounding mechanism is functionally equivalent to segmentation masks. In summary, our contributions include:

- An interactive object-aware audio generation model that links sounds to user-selected visual objects via masks.
- A mechanism that replaces attention with segmentation masks at test time, allowing fine-grained control over which objects, and thus which sounds, are present in the generated audio.
- Empirical and theoretical validation demonstrating our model outperforms baselines in sound-object alignment and user controllability while maintaining audio quality.

## 2. Related Work

**Object discovery.** Object-centric learning aims to represent visual scenes as compositions of discrete objects, enabling models to understand and manipulate individual entities within a scene. Unsupervised object discovery methods have been developed to decompose visual scenes into object representations without explicit annotations (Greff et al., 2019; Burgess et al., 2019; Locatello et al., 2020). In the audio-visual realm, prior studies have explored audio localization (Arandjelovic & Zisserman, 2018; Rouditchenko et al., 2019; Chen et al., 2021b; Mo & Morgado, 2022; Hamilton et al., 2024), separation (Zhao et al., 2018; Afouras et al., 2020), and spatialization (Li et al., 2024b) by using the correspondence between visual objects and their corresponding audio. In concurrent work, SSV2A (Guo

et al., 2024) introduces bounding boxes from external object detectors to generate audio from multiple sound sources. In contrast, our model interactively generates object-specific sound, without requiring explicit object segmentations and representations during training.

**Predicting sound from images and text.** Generating sounds from visual and textual inputs has gained notable attention recently. Image-based methods focus on synthesizing sounds from visual cues such as physical interactions (Van Den Doel et al., 2001; Owens et al., 2016), human movements (Gan et al., 2020; Su et al., 2021; Ephrat & Peleg, 2017; Prajwal et al., 2020; Hu et al., 2021), musical instrument performances (Koepke et al., 2020), and content from open-domain images and videos (Zhou et al., 2018; Iashin & Rahtu, 2021; Sheffer & Adi, 2023; Luo et al., 2023; Tang et al., 2023; Wang et al., 2024; Tang et al., 2024; Xing et al., 2024; Zhang et al., 2024; Cheng et al., 2024a; Chen et al., 2024). These approaches typically generate audio that corresponds to the entire visual scene without isolating individual sound sources, resulting in holistic sound generation. Text-based methods aim to produce sounds from textual descriptions using generative models like GANs and diffusion models (Yang et al., 2023; Kreuk et al., 2023; Liu et al., 2023; Huang et al., 2023b; Saito et al., 2025; Evans et al., 2025). However, when prompts contain multiple sound events, these methods often struggle to capture all the desired audio elements (Wu et al., 2023). Unlike these models, our method generates sounds for user-selected one or more objects within images. This offers enhanced control and precision in audio generation.

**Audio-visual learning.** Many works have focused on audio-visual associations due to their inherent correspondence in videos. A line of works explores the semantic

correspondence, identifying which sounds and visuals are commonly associated with one another (Arandjelovic & Zisserman, 2017). This includes representation learning (Morgado et al., 2021; Huang et al., 2023a), source localization (Chen et al., 2021b; Harwath et al., 2018; Chen et al., 2023b), audio stylization (Chen et al., 2022a; Li et al., 2024a), as well as scene classification (Chen et al., 2020; Gemmeke et al., 2017; Du et al., 2023a) and generation (Li et al., 2022b; Sung-Bin et al., 2023). Other studies leverage spatial correspondence between audio and visual streams (Owens & Efros, 2018; Korbar et al., 2018; Patrick et al., 2021) to tackle tasks like source separation (Zhao et al., 2018; 2019; Ephrat et al., 2016; Gao et al., 2018; Li et al., 2020; Chen et al., 2023a; Dong et al., 2022; Cheng et al., 2024b), Foley sound synthesis (Owens et al., 2016; Du et al., 2023b), and audio spatialization (Gao & Grauman, 2019; Morgado et al., 2018; Yang et al., 2020). Inspired by these works, we aim to generate sound from user-selected objects within images.

# 3. Interactive Object-Aware Audio Generation

Our goal is to generate sound from user-selected objects within an image in an interactive way. We cast this problem by learning the correlation between audio and its corresponding visual scene and then using this correlation to predict the sound from the activated region. To achieve this, we: (i) fine-tune an off-the-shelf conditional audio generation model for sound synthesis; (ii) train an audio-guided visual object grounding model to isolate the desired object; (iii) theoretically demonstrate the equivalence between the segmentation mask and our grounding model.

## 3.1. Conditional Audio Generation Model

**Conditional latent diffusion model.** We adopt a pre-trained conditional latent diffusion model (Liu et al., 2023) to generate audio conditioned on textual inputs. Building upon latent diffusion models (Ho et al., 2020; Rombach et al., 2022), our model operates in the latent space to improve computational efficiency. Specifically, given a text prompt $\boldsymbol{t}_q$ describing the desired sound and a noise vector $\boldsymbol{\epsilon} \sim \mathcal{N}(\mathbf{0}, \mathbf{I})$, the model iteratively denoises the latent variables over $N$ steps to generate the corresponding audio.

Our model is trained to predict the added noise at each denoising step $n$, conditioned on the textual input $\boldsymbol{t}_q$. The training objective minimizes the difference between the predicted noise and the true noise:

$$\mathcal{L}_\theta = \mathbb{E}_{\boldsymbol{z}_0, \boldsymbol{t}_q, \boldsymbol{\epsilon} \sim \mathcal{N}(\mathbf{0}, \mathbf{I}), n} \|\boldsymbol{\epsilon} - \boldsymbol{\epsilon}_\theta(\boldsymbol{z}_n, n, \boldsymbol{t}_q)\|_2^2, \quad (1)$$

where $\boldsymbol{z}_0$ is the latent representation of the ground truth audio, $\boldsymbol{z}_n$ is the noisy latent at step $n$, and $\boldsymbol{\epsilon}_\theta$ is the denoising model parameterized by $\theta$.

**Mel-spectrograms compression.** We compress mel-spectrograms into a lower-dimensional latent space using a variational autoencoder (VAE) (Kingma & Welling, 2013). The VAE encodes the mel-spectrogram $\boldsymbol{a} \in \mathbb{R}^{T \times F}$ into a latent representation $\boldsymbol{z} \in \mathbb{R}^{T' \times F' \times d}$, where $T'$ and $F'$ are reduced temporal and frequency dimensions, and $d$ is the dimensionality of the latent embeddings.

**Textual representation.** We represent the textual input $\boldsymbol{t}_q$ using a pre-trained text encoder from CLAP (Elizalde et al., 2023), which maps the text into an embedding space $\mathcal{E}_t(\boldsymbol{t}_q) \in \mathbb{R}^L$, where $L$ denotes the embedding dimension. These text embeddings capture semantic information about the desired sound and are used to condition the diffusion model through cross-attention mechanisms (Vaswani et al., 2017).

**Classifier-free guidance.** We employ classifier-free guidance (CFG) (Ho & Salimans, 2022) to encourage the model to learn both conditional and unconditional denoising. During training, we randomly omit the conditioning input $\boldsymbol{t}_q$ with a 10% probability. At test time, we use a guidance scale $\lambda \geq 1$ to interpolate between the conditional and unconditional predictions:

$$\tilde{\boldsymbol{\epsilon}}_\theta(\boldsymbol{z}_n, n, \boldsymbol{t}_q) = \lambda \cdot \boldsymbol{\epsilon}_\theta(\boldsymbol{z}_n, n, \boldsymbol{t}_q) + (1-\lambda) \cdot \boldsymbol{\epsilon}_\theta(\boldsymbol{z}_n, n, \emptyset), \quad (2)$$

where $\boldsymbol{\epsilon}_\theta(\boldsymbol{z}_n, n, \emptyset)$ is the unconditional prediction.

**Waveform reconstruction.** After generating the latent representation of the audio, we reconstruct the corresponding waveform. The decoder part of the VAE transforms the latent representation $\boldsymbol{z}_0$ back into a mel-spectrogram. Subsequently, a pre-trained HiFi-GAN neural vocoder (Kong et al., 2020a) is used to synthesize the time-domain audio waveform from the mel-spectrogram, producing the final audio output.

## 3.2. Text-Guided Visual Object Grounding Model

**Visual representation.** To ground the visual objects corresponding to the desired sound, we extract features from the input image using a pre-trained visual encoder. Specifically, we utilize CLIP (Radford et al., 2021) to encode the image into a set of visual patches embeddings $\mathcal{E}_v(\boldsymbol{i}_q) \in \mathbb{R}^{P \times L}$, where $\boldsymbol{i}_q$ is the input image, $P$ is the number of patches, and $L$ denotes the embedding dimension (matching that of the text embeddings). These embeddings capture both semantic and spatial information of the visual scene.

**Scaled dot-product attention.** We employ scaled dot-product attention (Vaswani et al., 2017) to fuse the textual and visual inputs, allowing the model to focus on specific objects within the scene. Before computing the attention, the text embeddings $\mathcal{E}_t(\boldsymbol{t}_q)$ and patch embeddings $\mathcal{E}_v(\boldsymbol{i}_q)$

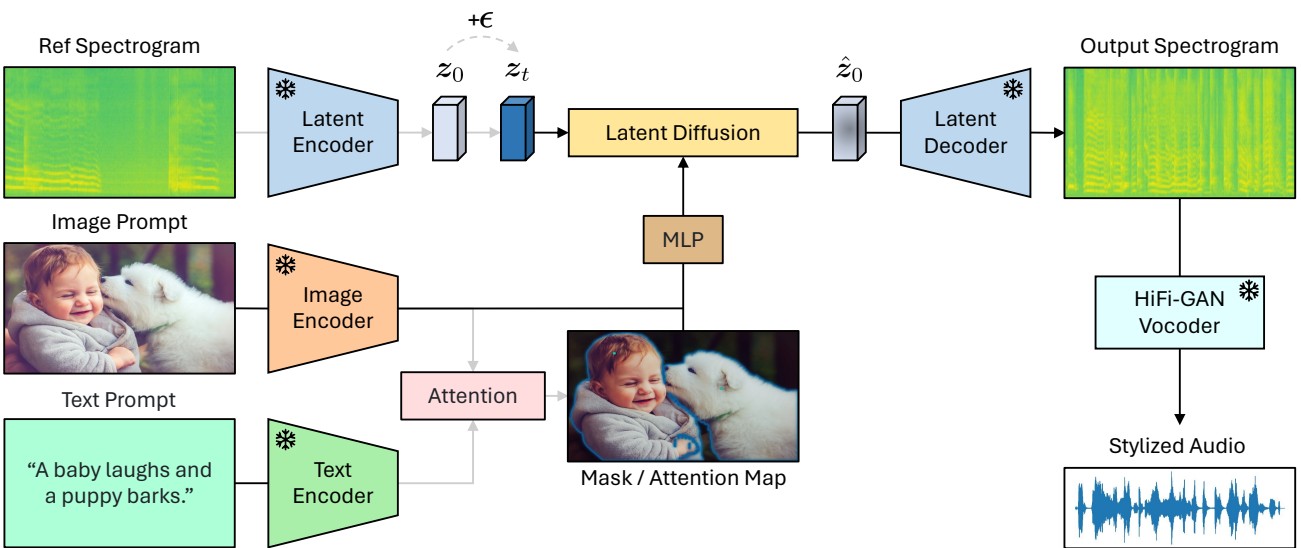

Figure 2: **Model architecture.** We encode the reference spectrogram via a pre-trained latent encoder. An image and text prompt are processed by separate encoders, and their embeddings are fused using an attention mechanism to highlight relevant objects. We then feed these conditioned features and noisy latent into a latent diffusion model to generate the object-specific audio. Finally, the latent decoder reconstructs the spectrogram, and a pre-trained HiFi-GAN vocoder generates the final audio waveform. At test time, we replace the attention with a user-provided segmentation mask, and the latent encoder for the reference spectrogram is *not* used.

are linearly projected to obtain the query, key, and value matrices. Specifically, we compute:

$$\boldsymbol{Q} = \mathcal{E}_t(\boldsymbol{t}_q)\boldsymbol{W}^Q, \ \boldsymbol{K} = \mathcal{E}_v(\boldsymbol{i}_q)\boldsymbol{W}^K, \ \boldsymbol{V} = \mathcal{E}_v(\boldsymbol{i}_q)\boldsymbol{W}^V, \tag{3}$$

where $\boldsymbol{W}^Q$, $\boldsymbol{W}^K$, and $\boldsymbol{W}^V$ are learnable projectors.

We then compute the attention weights between the projected text and each projected image patch, grounding the text in the visual domain:

$$\text{Attention}(\boldsymbol{Q}, \boldsymbol{K}, \boldsymbol{V}) = \text{softmax}\left(\frac{\boldsymbol{Q}\boldsymbol{K}^\top}{\sqrt{d_k}}\right)\boldsymbol{V}, \tag{4}$$

where $d_k$ is the dimensionality of the key embedding.

After obtaining the attention output, we apply an MLP layer (Murtagh, 1991) to further refine the fused representations, which enables the model to attend to image regions corresponding to the text input. In this way, we integrate the images $\boldsymbol{i}_q$ with the diffusion process, allowing the model to learn to focus on the relevant regions in the image through self-supervision.

**Learnable positional encoding.** To enhance the model's ability to localize objects within the image, we incorporate learnable positional encodings (Devlin, 2018) into the attention mechanism. These encodings are added to the key and value embeddings, providing spatial information about the image patches. By learning positional information, the model can better distinguish between objects in different locations, improving grounding precision.

**Segmentation mask at test time.** After training, we substitute the attention weights derived from the scaled dot-product attention with segmentation masks generated by the segment anything model (SAM) (Kirillov et al., 2023). We rescale the raw outputs of SAM into a normalized mask $\boldsymbol{m}_q \in \mathbb{R}^P$, matching the mean and variance of the attention weights. This allows us to generate the desired object's sound by focusing on the regions specified by the segmentation mask. Since SAM's masks can be obtained using either text prompts or point clicks, our model supports interactive image-to-audio generation, allowing users to intuitively select objects of interest and generate their associated sounds.

### 3.3. Theoretical Analysis

One may notice that our training pipeline uses both text and image encoders, but the test-time computation involves only the image encoder, where the softmax attention weights are replaced by the segmentation masks. This indicates an *out-of-distribution* generalization ability (Lin et al., 2023), where our model trained on the softmax attention weights computed by CLAP & CLIP embeddings (Equation 4) is able to generalize well on the segmentation masks computed by SAM. We hypothesize that this ability is rooted in the alignment of contrastive losses and the dot-product attention mechanism. Recall that the InfoNCE loss (Gutmann & Hyvärinen, 2010; Oord et al., 2018) for the text encoder in

contrastive learning is given by $\mathcal{L}_t \left( \mathcal{E}_t, \mathcal{E}_v \right) =:$

$$\mathbb{E}_{x^T, x^I_{1:N}} \left[ -\log \frac{\exp \left( \langle \mathcal{E}_v(x^T), \mathcal{E}_t(x^I_1) \rangle / \tau \right)}{\sum_{j=1}^{N} \exp \left( \langle \mathcal{E}_v(x^T), \mathcal{E}_t(x^I_j) \rangle / \tau \right)} \right] \quad (5)$$

where $(x^T, x^I_1)$ is the matching text-image pair, and $x^I_2, \ldots, x^I_N$ are the negative image samples associated with $x^T$. Notice that if we substitute $x^T$ with the text input $\boldsymbol{t}_q$, $x^I_{1:N}$ with the image patches $\boldsymbol{i}_q$, and $x^I_1$ with the matching image patch (with the text input), then the loss in Equation 5 becomes the Maximum Likelihood Estimation (MLE) loss of the softmax attention weights in Equation 4 (under proper scaling in the exponents). Therefore, the encoders $\mathcal{E}_v, \mathcal{E}_t$ are able to assign high attention weights to image patches that match with textual inputs, and low attention weights to irrelevant image patches, *working effectively as the segmentation mask at test time*. As such, the audio generation model is trained with the ability to focus only on the selected objects by segmentation masks.

In the following theorem, we formalize the above argument into a test-time error guarantee. We let $f$ denote the composition of the trained MLP layers and the trained audio generation model, such that $f$ maps an attention output $a_q$ to an audio output $s_q$ (on query $q$), and let $v$ denote the value metric that maps a sound-image-mask tuple $(s, i, m)$ to a real number $v(s, i, m) \in \mathbb{R}$. Our goal is to bound the following test error

$$\text{err}_{\text{test}} := \mathbb{E}_q[v(f^*(p_q \boldsymbol{V}^*), \boldsymbol{i}_q, p_q) - v(f(\boldsymbol{m}_q \boldsymbol{V}), \boldsymbol{i}_q, \boldsymbol{m}_q)]$$

i.e. the expected (over the randomness of test query $q$) gap between (i) the value $v(f^*(p_q \boldsymbol{V}^*), \boldsymbol{i}_q, p_q)$ achieved by the optimal model $(f^*, \boldsymbol{V}^*)$ and ground-truth mask $p_q$, and (ii) the value $v(f(\boldsymbol{m}_q \boldsymbol{V}), \boldsymbol{i}_q, \boldsymbol{m}_q)$ of the trained model $(f, \boldsymbol{V})$ using SAM segmentation $\boldsymbol{m}_q$ at test time. Here, $f^*$ and $\boldsymbol{V}^*$ are the ground-truth counterpart of $f$ and value matrix, $p_q \in \Delta^P$ is the (normalized) ground-truth mask of query $q$ such that $p_{q,k} = \frac{\mathbb{P}(\boldsymbol{t}_q | \boldsymbol{i}_{q,k})}{\sum_{l=1}^{P} \mathbb{P}(\boldsymbol{t}_q | \boldsymbol{i}_{q,l})}$ for patch index $k \in \{1, \ldots, P\}$, $a_q$ represents the attention output computed by Equation 4. Note that $f(\boldsymbol{m}_q \boldsymbol{V})$, the audio output of the trained model, depends on the segmentation mask $\boldsymbol{m}_q$ instead of the ground-truth mask $p_q$ or text input $\boldsymbol{t}_q$.

**Theorem 3.1.** *Let $\epsilon_{\text{sam}} := \mathbb{E}_q[\|\boldsymbol{m}_q - p_q\|_{\ell_1}]$ denote the expected $\ell_1$ error of the segmentation model. Let $\epsilon_f, \epsilon_{\boldsymbol{V}}$ denote the expected error of $f$ and $\boldsymbol{V}$ under the pre-trained CLAP & CLIP embeddings respectively, and $\epsilon_{\text{contrast}}$ denote the expected contrastive loss of the encoders, more precisely,*

$$\epsilon_f = \mathbb{E}_q[v(f^*(a_q), \boldsymbol{i}_q, p_q)] - \mathbb{E}[v(f(a_q), \boldsymbol{i}_q, p_q)],$$
$$\epsilon_{\boldsymbol{V}} = \|\boldsymbol{V} - \boldsymbol{V}^*\|_\infty$$

$$\epsilon_{\text{contrast}} = \mathbb{E}_{q, d \sim p_q} \left[ -\log \frac{\exp \left( \langle \mathcal{E}_v(\boldsymbol{t}_q), \mathcal{E}_t(i_{q,d}) \rangle_\Sigma \right)}{\sum_{k=1}^{P} \exp \left( \langle \mathcal{E}_v(\boldsymbol{t}_q), \mathcal{E}_t(i_{q,k}) \rangle_\Sigma \right)} \right]$$
$$- \mathbb{E}_{q, d \sim p_q} \left[ -\log p_{q,d} \right].$$

*where $\langle \cdot, \cdot \rangle_\Sigma$ is the local inner product under $\Sigma := \boldsymbol{W}^K (\boldsymbol{W}^Q)^\top / \sqrt{d_k}$ (note that $\epsilon_{\text{contrast}}$ is simply the difference between the model's InfoNCE loss and the optimal InfoNCE loss, under the similarity metric $\langle \cdot, \cdot \rangle_\Sigma$). Suppose $\|\boldsymbol{V}^*\|_\infty, \|\boldsymbol{V}\|_\infty \leq B_v$, $v$ is $L_v$-Lipschitz, and $f, f^*$ are $L_f$-Lipschitz, then we have*

$$\text{err}_{\text{test}} \leq L_v \cdot L_f \cdot \left( \epsilon_{\boldsymbol{V}} + B_v \cdot \left( \epsilon_{\text{sam}} + 2\sqrt{2\epsilon_{\text{contrast}}} \right) \right)$$
$$+ L_v \cdot \epsilon_{\text{sam}} + \epsilon_f. \quad (6)$$

Due to space constraints, the proof is deferred to the Appendix F. On the right hand side of Equation 6, the error terms $\epsilon_{\boldsymbol{V}}, \epsilon_{\text{sam}}, \epsilon_{\text{contrast}}, \epsilon_f$ have been minimized by massive training (Radford et al., 2021; Elizalde et al., 2023; Kirillov et al., 2023); furthermore, the regularity parameters $L_v, L_f, B_v$ are standard in learning theory literature (Anthony & Bartlett, 1999; Neyshabur et al., 2015; Bartlett et al., 2017) and can be bounded with guarantees (Tsuzuku et al., 2018; Combettes & Pesquet, 2020; Fazlyab et al., 2019). Consequently, Theorem 3.1 implies that the test-time error can be effectively upper bounded, hence supporting the substitution of attention weights derived from scaled dot-product attention with segmentation masks generated by the segmentation model during testing. Our theory is further corroborated by empirical findings in Section 4.3, where we observe that using dot-product attention weights achieves performance on par with using segmentation masks, while additive attention fails completely.

## 4. Experiments

### 4.1. Experiment Setup

**Dataset.** We use AudioSet (Gemmeke et al., 2017) as our primary data source, which consists of 4,616 hours of video clips, each paired with corresponding labels and captions. To ensure audio-visual correspondence, we perform several preprocessing steps similar to Sound-VECaps (Yuan et al., 2024). This reduces the dataset to 748 hours of video for training. We then evaluate models on the AudioCaps (also a subset of AudioSet) (Kim et al., 2019), a widely used benchmark dataset for audio generation. Please see Appendix B for more details on the dataset.

**Model architecture.** We employ the pre-trained VAE and HiFi-GAN vocoder in AudioLDM (Liu et al., 2023). The VAE is configured with a latent dimensionality $d$ of 8 channels. For embedding extraction, we utilize the "ViT-B/32" CLAP audio encoder (Elizalde et al., 2023) and the CLIP image encoder (Radford et al., 2021). These embeddings are then incorporated into the U-Net-based diffusion model through cross-attention. We implement a linear noise schedule consisting of $N = 1000$ diffusion steps, from $\beta_1 = 0.0015$ to $\beta_N = 0.0195$. The DDIM sampling

| Method | ACC (↑) | FAD (↓) | KL (↓) | IS (↑) | AVC (↑) | OVL (↑) | RET (↑) | REI (↑) | REO (↑) |
|---|---|---|---|---|---|---|---|---|---|
| Ground Truth | / | / | / | / | 0.962 | 4.12 ± 0.06 | 4.02 ± 0.05 | 4.06 ± 0.07 | / |
| Retrieve & Separate (Zhao et al., 2018) | 0.276 | 4.051 | 1.572 | 1.550 | 0.764 | 2.73 ± 0.02 | 2.54 ± 0.05 | 2.76 ± 0.04 | 2.49 ± 0.04 |
| AudioLDM 1 (Liu et al., 2023) | 0.336 | 3.576 | 1.537 | 1.545 | 0.724 | 2.83 ± 0.07 | 3.09 ± 0.03 | 2.92 ± 0.02 | 2.18 ± 0.04 |
| AudioLDM 2 (Liu et al., 2024) | 0.513 | 2.976 | 1.162 | 1.779 | 0.743 | 2.98 ± 0.04 | 3.19 ± 0.02 | 3.09 ± 0.03 | 2.47 ± 0.01 |
| Captioning (Li et al., 2022a) | 0.587 | 2.778 | 1.364 | 1.901 | 0.773 | 2.84 ± 0.03 | 3.15 ± 0.04 | 3.05 ± 0.06 | 2.63 ± 0.05 |
| Make-an-Audio (Huang et al., 2023b) | 0.309 | 3.555 | 1.443 | 1.673 | 0.712 | 2.74 ± 0.08 | 3.06 ± 0.05 | 2.89 ± 0.05 | 2.08 ± 0.04 |
| Im2Wav (Sheffer & Adi, 2023) | 0.499 | 3.602 | 1.526 | 1.872 | 0.798 | 2.88 ± 0.05 | 3.12 ± 0.04 | 3.01 ± 0.05 | 2.48 ± 0.06 |
| SpecVQGAN (Iashin & Rahtu, 2021) | 0.611 | 2.515 | 1.142 | 1.965 | 0.825 | 2.94 ± 0.04 | 3.26 ± 0.03 | 3.11 ± 0.06 | 2.51 ± 0.04 |
| Diff-Foley (Luo et al., 2023) | 0.683 | 1.908 | 0.783 | 2.010 | 0.842 | 3.09 ± 0.06 | 3.43 ± 0.05 | 3.32 ± 0.03 | 2.52 ± 0.06 |
| CoDi (Tang et al., 2023) | 0.672 | 1.954 | 0.856 | 1.936 | 0.833 | 3.00 ± 0.04 | 3.32 ± 0.03 | 3.31 ± 0.05 | 2.34 ± 0.02 |
| Seeing & Hearing (Xing et al., 2024) | 0.668 | 1.923 | 0.794 | 1.954 | 0.722 | 3.08 ± 0.05 | 3.38 ± 0.04 | 3.28 ± 0.06 | 2.49 ± 0.04 |
| FoleyCrafter (Zhang et al., 2024) | 0.732 | 1.760 | 0.665 | 2.007 | 0.811 | 3.19 ± 0.02 | 3.48 ± 0.03 | 3.32 ± 0.04 | 2.60 ± 0.04 |
| SSV2A (Guo et al., 2024) | 0.806 | **1.265** | 0.525 | 2.100 | 0.893 | 3.22 ± 0.02 | 3.50 ± 0.03 | 3.35 ± 0.02 | 3.48 ± 0.06 |
| Ours | **0.859** | 1.271 | **0.517** | 2.102 | 0.891 | **3.31 ± 0.04** | **3.62 ± 0.05** | **3.48 ± 0.04** | **3.74 ± 0.07** |

Table 1: Quantitative comparison of our method and baselines across different objective and subjective metrics. The subjective OVL, RET, REI, and REO scores are presented with 95% confidence intervals.

method (Song et al., 2020) is used with 200 steps to facilitate efficient generation. At test time, we apply CFG with a guidance scale $\lambda$ set to 2.0.

**Training configuration.** To facilitate parallel training, each video's soundtrack is either truncated or zero-padded to achieve a fixed duration of 10 seconds and then converted to a 16 kHz sample rate. We apply a 512-point discrete Fourier transform with a frame length of 64 ms and a frame shift of 10 ms. For each video, a single visual frame is randomly chosen to serve as the input image. The model is then trained using the AdamW optimizer (Loshchilov & Hutter, 2017) with a batch size of 64, a learning rate of $10^{-4}$, $\beta_1 = 0.95$, $\beta_2 = 0.999$, $\epsilon = 10^{-6}$, and a weight decay of $10^{-3}$ over 300 epochs.

**Evaluation metrics.** We use both objective and subjective metrics (see Appendix C for more evaluation details) to evaluate the performance of our model. For the objective evaluation, we employ several metrics, including Sound Event Accuracy (ACC), which leverages the PANNs model (Kong et al., 2020b) to predict and sample sound event logits based on the annotated labels and then compute the mean accuracy across the dataset. We also measure the semantic alignment between the output and target using four established metrics: (i) Fréchet Audio Distance (FAD) (Kilgour et al., 2019), which quantifies how close the generated audio is to the real audio in latent space; (ii) Kullback-Leibler Divergence (KL), which assesses the alignment of distributions between the generated and target audio; (iii) Inception Score (IS) (Salimans et al., 2016), which evaluates the diversity of the generated audio; (iv) Audio-Visual Correspondence (AVC) (Arandjelovic & Zisserman, 2017), which measures how well the resulting audio match the visual context.

For subjective evaluation, we conduct a human study to assess the quality and relevance of the generated audio. We present both the holistic samples and the object-selected

samples. Each participant is provided with an input image, along with the corresponding generated audio, and is asked to rate each sample on a scale from 1 to 5 based on several criteria: (i) Overall Quality (OVL), which evaluates the general quality of the audio; (ii) Relevance to the Text Prompt (RET), which assesses how well the audio matches any associated text description; (iii) Relevance to the Input Image (REI), which judges the alignment between the audio and the visual content; (iv) Relevance to the Selected Object (REO), which focuses on how well the generated audio aligns with a specific object in the visual scene.

**Baselines.** We compare our method with several baseline models, each of which is adapted for our task: (i) Retrieve & Separate (Zhao et al., 2018), a two-stage object-aware model that first retrieves audio based on a text prompt (Elizalde et al., 2023), then separates the object-specific audio from the specified visual object (Zhao et al., 2018); (ii) AudioLDM 1 & 2 (Liu et al., 2023; 2024), which we fine-tune on our dataset for a fair comparison; (iii) Captioning (Li et al., 2022a), a cascade model that takes input image, generates captions and feeds them to a pre-trained AudioLDM 2; (iv) Make-an-Audio (Huang et al., 2023b), which supports either text or image prompts for audio generation. We extract its image branch and fine-tune it on our dataset; (v) Im2Wav (Sheffer & Adi, 2023), an image-guided open-domain audio generation model that operates auto-regressively. Since the original model generates only 4 seconds of audio, we retrain it on our dataset to better suit our task; (vi) SSV2A and (vii) CoDi, which are sound-source-aware and any-to-any generative models respectively. We use their image-to-audio branch for comparison; (viii) SpecVQGAN (Iashin & Rahtu, 2021), (ix) Diff-Foley (Luo et al., 2023), (x) Seeing & Hearing (Xing et al., 2024), (xi) FoleyCrafter (Zhang et al., 2024), which are video-to-audio generative models. We modify them by using static images (randomly sampling a single frame from each video clip) as input and fine-tuning them on our dataset.

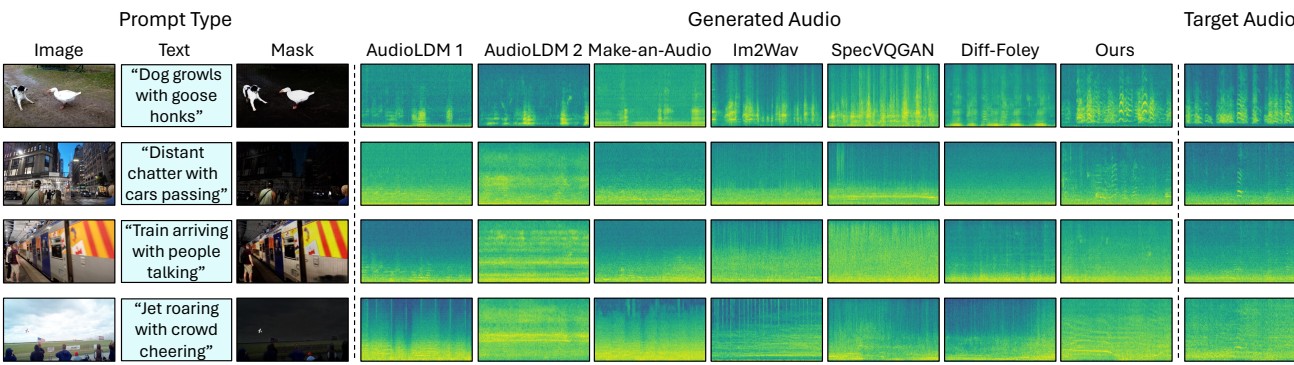

Figure 3: **Qualitative model comparison**. We show audio generation results for our method and the baselines, each of which is conditioned on an image, text, or segmentation mask.

## 4.2. Comparison to Baselines

**Quantitative results.** Table 1 compares our approach to baselines on the AudioCaps dataset (see Appendix E for evaluations on another dataset), showing that our model outperforms across metrics and generates high-quality audio. In particular, our method achieves the highest ACC scores, demonstrating its ability to generate sounds closely linked to visual objects in the scene. Among baselines, SSV2A performs competitively, likely due to its object-level specificity from the external object detector. Diff-Foley, Seeing & Hearing, and FoleyCrafter perform competitively, likely due to their contrastive representations enhancing audio-visual consistency. Make-an-Audio, Im2Wav, and SpecVQGAN achieve reasonable AVC scores but underperform on FAD and KL, suggesting limitations in audio quality. AudioLDM, Captioning, and CoDi show lower ACC and FAD metrics, likely reflecting that CLAP text embeddings fail to represent complex audio events. Retrieve & Separate struggles with retrieval in multi-source scenes, limiting its performance in complex scenarios. These results demonstrate our model's strength in leveraging object-level cues to generate contextually relevant sounds.

For subjective evaluation, we randomly select 100 generated samples from the test set, including 50 with manually created segmentation masks for specific objects (see Appendix C). These samples are then rated by 50 participants. Our model achieves the highest average ratings across all measures, with a significant lead in REO, indicating better alignment between generated sounds and objects in the image. Interestingly, baselines achieve similar REO scores, suggesting limited ability to link audio to object-level visual cues. Moreover, our model consistently outperforms in OVL, RET, and REI, further validating the objective metrics and demonstrating improved contextual alignment.

**Qualitative results.** Figure 3 compares our method with generative baselines on the AudioCaps dataset. In the first

| Method | Time (↓) | Attempts (↓) | Satisfaction (↑) |
|---|---|---|---|
| AudioLDM 1 (Liu et al., 2023) | 7.34 | 3.20 | 2.00 ± 0.88 |
| AudioLDM 2 (Liu et al., 2024) | 5.10 | 2.40 | 2.80 ± 1.04 |
| FoleyCrafter (Zhang et al., 2024) | 3.00 | 2.80 | 3.00 ± 1.96 |
| SSV2A (Guo et al., 2024) | 2.95 | 1.80 | 3.40 ± 1.42 |
| Ours | **2.67** | **1.60** | **3.60 ± 0.68** |

Table 2: Interaction satisfaction evaluation of user-driven audio generation methods. We report average time (minutes), number of attempts, and satisfaction score (with 95% confidence intervals).

example, where a dog and a goose are present, baselines generate only dog growls, missing the goose honks, while our method captures both sounds, demonstrating its object-aware capability. Similarly, in the second and third examples, baselines produce only partial sound events, whereas our model generates the complete soundscape. In the final example, featuring a small jet in the background with a cheering crowd, vision-based models fail to detect the jet due to its small size, generating only crowd and wind noises, while text-based models struggle to combine multiple sounds. Our approach captures all relevant sounds, highlighting its ability to generate accurate audio aligned with complex visual scenes. For a more direct experience, please view the results video on the project webpage.

**Interaction satisfaction.** We conduct another human study focusing on user-driven audio generation, comparing our method to text-based baselines (we exclude those that do not allow user prompting). We ask 5 experienced participants to generate "baby laughs and puppy barks" from a single image (the one in Figure 2), and we measure the average time taken, the number of attempts required, and a 5-point subjective satisfaction score. As shown in Table 2, text-based baselines often miss one of the sounds and require multiple prompt adjustments, leading to higher time and lower satisfaction. Our method, by contrast, consistently requires fewer attempts, takes less time, and achieves

| Method | ACC (↑) | FAD (↓) | KL (↓) | IS (↑) | AVC (↑) |
|---|---|---|---|---|---|
| (i) Frozen Diffusion | 0.692 | 1.543 | 1.047 | 1.943 | 0.733 |
| (ii) Muiti-Head Attn. | 0.415 | 2.238 | 1.903 | **2.115** | 0.887 |
| (iii) Additive Attn. | 0.103 | 15.747 | 7.425 | 1.343 | 0.137 |
| (iv) Txt-Img Attn. | 0.856 | **1.270** | 0.520 | 2.097 | 0.890 |
| (v) Aud-Img Attn. | 0.634 | 1.761 | 1.232 | 1.731 | 0.692 |
| (vi) Mask Training | 0.763 | 1.446 | 0.742 | 1.947 | 0.797 |
| Ours | **0.859** | 1.271 | **0.517** | 2.102 | **0.891** |

Table 3: Quantitative ablation studies on the AudioCaps dataset.

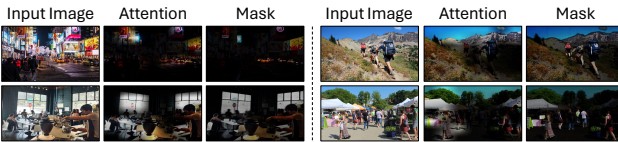

Figure 4: **Visualization results**. We visualize the difference between attention maps and segmentation masks using images from Places (Zhou et al., 2017) and text prompts from BLIP (Li et al., 2022a).

higher satisfaction, even for participants already familiar with prompting.

### 4.3. Ablation Study and Analysis

Table 3 summarizes the ablation experiments. We explore the following model variations: (i) freezing the latent diffusion weights rather than fine-tuning them; (ii) replacing single-head attention with multi-head attention; (iii) altering the attention mechanism from dot-product to additive attention; (iv) using text-image attention instead of segmentation masks during inference; (v) substituting text-image attention with audio-image attention; (vi) using segmentation masks instead of attention during training. We also show additional results in Appendix D.

**Effect of freezing diffusion weights.** We test the impact of freezing the latent diffusion model weights instead of fine-tuning them during training. We observe that freezing the weights degrades the performance, which suggests that fine-tuning is required to achieve more coherent audio.

**Impact of attention head.** We compare our single-head attention mechanism with the multi-head counterpart (Vaswani et al., 2017). The multi-head approach enhances the alignment between textual inputs and the generated audio, leading to a stronger correspondence between text descriptions and sound outputs. However, this improvement reduces controllability when specifying specific audio characteristics based on the segmentation mask. We conjecture that this limitation arises because each head in the multi-head attention focuses on different regions of the input (Voita et al., 2019; Hamilton et al., 2024). While this strategy increases text-audio alignment, the lack of a clear definition for each head's specific scope reduces the interpretability of the final results. This likely contributes to the masking results deviating from expectations.

**Evaluation of attention scoring mechanism.** We assess the role of the attention scoring function by replacing dot-product attention with the additive one (Bahdanau, 2014). The additive variant collapses significantly, indicating that segmentation masks are not a suitable replacement for this attention. As explained by the theory in Section 3.3, this could be because addition operations are not compatible with the contrastive losses used by CLAP & CLIP and segmentation masks generated by SAM, which disrupts our grounding model.

**Choice of attention modality.** We investigate the effectiveness of text-image attention compared to an adapted audio-image attention model (Li et al., 2024b). Results show a decline in performance, which could be attributed to the inherent limitations of the CLAP model in representing overlapping audio. This limitation probably introduces noise, thereby weakening the model's ability to form audio-visual associations essential for audio generation.

**Role of masking during training and inference.** We compare the text-image attention to segmentation masks at test time. Results show that this attention achieves comparable performance to segmentation masks, suggesting both methods provide similar guidance (Section 3.3). Notably, using segmentation masks in both training and testing degrades performance. We hypothesize that masking entire object regions imposes an overly rigid prior, as sound is typically emitted from specific parts (e.g., a dog's head rather than its tail). From a probabilistic viewpoint, hard masks sampled from the ground-truth distribution exhibit high variance, whereas soft attention, empowered by CLIP & CLAP, directly approximates the ground-truth distribution. This allows the model to focus on sound-relevant regions while maintaining audio accuracy at test time.

### 4.4. Cross-dataset Evaluation

**Visualization between grounding and masking.** In Figure 4, we visualize the comparison between the attention maps generated by our model and the segmentation masks produced by SAM. For this, we use images from Places (Zhou et al., 2017) and text prompts derived from BLIP (Li et al., 2022a). To visualize the attention maps, we apply bilinear interpolation to match the resolution of the segmentation masks. Our results show a strong alignment between our model's attention maps and the segmentation masks, providing empirical evidence for the theory in Section 3.3 and the findings of the ablation study in Section 4.3. While the segmentation masks represent a form of *hard* attention,

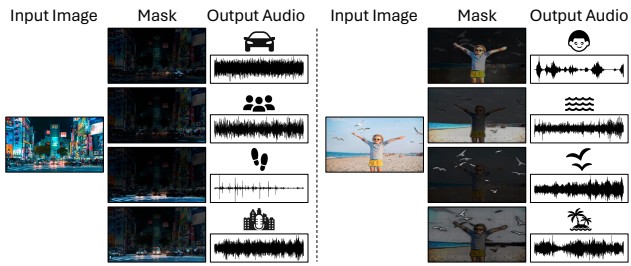

Figure 5: **Interactive audio generation**. Our model generates object-specific sounds in the city (left) and beach (right) scenes, and composes a complete soundscape when one or more objects are selected.

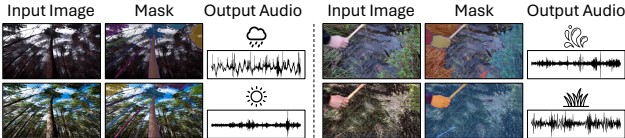

Figure 6: **Generating soundscapes from visual texture changes.**. We generate different soundscapes by manipulating the visual textures of the same scene, such as changing weather (left) or materials (right).

directly highlighting entire selected objects, our model generates *soft* attention maps that probabilistically focus on the sound-relevant areas within each object. This similarity indicates that, through training, our model learns to capture object-specific regions similar to those identified by segmentation, achieving the desired grounding in a flexible manner. Furthermore, this observation suggests that attention maps can be replaced with segmentation masks at test time.

**Interactive audio generation.** We ask whether our model will generate object-specific sounds by isolating individual objects within a scene. As shown in Figure 5, we use the same image for each scene, separating different objects (cars, people, seagulls, etc.) to generate corresponding audio outputs. The results illustrate that our model successfully learns to generate distinct sounds for each object, such as car engines or footsteps, reflecting their unique sound textures. Furthermore, when multiple objects are selected together, our model is able to generate the entire soundscape that represents the scene property. This capability highlights our model's strength in interactively synthesizing audio.

**Sound adaptation to visual texture changes.** We explore whether our method can generate soundscapes that adapt to changes in visual textures, inspired by audio-visual video editing (Lee et al., 2023). Starting with images from the Places (Zhou et al., 2017) and Greatest Hits (Owens et al., 2016) datasets, we apply an off-the-shelf image translation model (Park et al., 2020; Li et al., 2022b) to create paired scenes (e.g., sunny-rainy, water-grass), and then overlay full-

image segmentation masks on top. As illustrated in Figure 6, our model generates context-appropriate soundscapes. For instance, it generates rain sounds for dark skies, wind sounds for clear skies, water splashing for watery surfaces, and grass crunching for grassy areas. This demonstrates that our model successfully captures variations in visual textures to generate corresponding audio.

**Balancing the volume of different objects.** We find in Figure 5 that specifying each object separately tends to assign a similar volume to all sources. However, when multiple objects are selected, our method dynamically accounts for context. For example, if a large car dominates the scene, its siren may overwhelm subtle ambient sounds, creating a more realistic blend instead of flattening everything to equal volume. Moreover, we quantitatively confirm this context-driven behavior in Table 1, 7, and 10, where our object-aware method better reflects how certain sources can overpower others or combine to create natural audio events.

**Interactions among multiple objects.** We show in Figure 6 that our method captures interactions, like a stick splashing water, instead of generating only generic water flowing sounds. These results indicate our model's ability to handle basic multi-object interactions from static images.

## 5. Conclusion

In this paper, we proposed an *interactive object-aware audio generation* model, focusing on aligning generated sounds with specific visual objects in complex scenes. To achieve this, we developed a diffusion model grounded in object-centric representations, enhancing the association between objects and their corresponding sounds via multi-modal attention. Theoretical analysis demonstrates that our object-grounding mechanism is functionally equivalent to segmentation masks. Quantitative and qualitative evaluations show that our model surpasses baselines in sound-object alignment, enabling cross-dataset generalization and user-controllable synthesis. We hope our work not only advances controllable audio generation but also inspires further exploration into the relationships between objects and sounds. We will release code and models upon acceptance.

**Limitations and broader impacts.** Our model shows promising results in generating object-specific sounds from images but has certain limitations. First, relying on static images makes it challenging to produce non-stationary audio synchronized with dynamic events, such as impact sounds (Figure 6). Second, it may lack precise control over the type of sound generated for similar objects, leading to ambiguity (e.g., a car might produce a siren or engine noise in Figure 3). Lastly, while useful for content creation like filmmaking, our model could be misused to generate misleading videos.

## Acknowledgment

We thank Ziyang Chen, Hao-Wen Dong, Yisi Liu, and Zhikang Dong for their helpful discussions, and the anonymous reviewers for their valuable feedback.

## Impact Statement

This paper introduces an *interactive object-aware audio generation* model. It is trained on a publicly available dataset, i.e., AudioSet, which does not contain personally identifiable information. We have taken steps to ensure compliance with data usage policies, and our model does not involve human subjects or raise privacy concerns. We believe our work poses minimal negative ethical impacts and societal implications, as it focuses on enhancing sound-object alignment in a controlled research environment. However, we encourage responsible use of our model, particularly when applied to real-world scenarios.

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

## A. Results Video

We provide a results video on the project webpage, which showcases our model's ability to generate sounds based on the masked object prompts. Specifically, this video demonstrates the following:

- Our model can interactively generate object-specific sounds within complex scenes.
- Despite being trained on the AudioSet (Gemmeke et al., 2017), our model can be successfully applied to out-of-domain visual scenes, including those from the Places dataset (Zhou et al., 2017), the Greatest Hits dataset (Owens et al., 2016), and even random web images.
- Our model can capture variations in visual textures to generate corresponding audio.
- Our model can capture diverse objects within an image and generate sounds more accurately than the baselines.

## B. Dataset Preprocessing Details

### B.1. Dataset Refinement

We use the AudioSet (Gemmeke et al., 2017) as the primary source for this task. The original dataset comprises 4,616 hours of video clips, each paired with corresponding labels and captions. Inspired by Sound-VECaps (Yuan et al., 2024), we apply the following refinement steps to adapt the dataset for our use.

**Audio-visual matching.** To ensure strong correspondence between audio and visual inputs, we train an audio-visual matching model (Figure 8), which consists of a 6-layer non-causal transformer with a rotary positional embedding mechanism (Su et al., 2024). Visual embeddings are extracted using the ViT-B/16 Transformer module from CLIP (Radford et al., 2021), while audio embeddings are generated using the BEATs model (Chen et al., 2022b). Both embeddings are then passed through a 3-layer MLP to match a 768-dimensional space. The model is trained in a self-supervised manner (Owens & Efros, 2018; Korbar et al., 2018), treating audio-visual pairs from the same temporal instance as matches and those from different videos as mismatches, which allows the model to learn audio-visual correspondences without human annotations.

For training efficiency, the videos are standardized to 8 frames per second, with each frame resized to 224x224 pixels. During the evaluation, our model achieves an accuracy of 91% for matching scenarios and 85% for non-matching scenarios on a set of 100 matched and 100 mismatched samples, indicating its effectiveness in capturing audio-visual alignment. We use this model to score each clip in the AudioSet, with results shown in Figure 7. A threshold of 0.6 is then applied to filter the dataset.

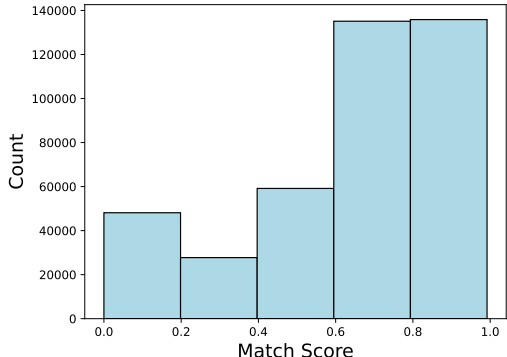

Figure 7: **Distribution of matching scores.** We present the scores for audio-visual pairs in the AudioSet.

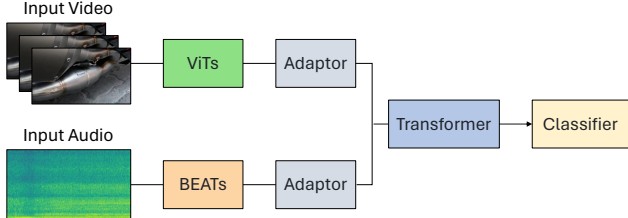

Figure 8: **Model architecture of audio-visual matching.** We train a model to quantify the correspondence between a video and its corresponding soundtrack.

**Caption rephrasing.** To ensure captions to focus exclusively on visible sounding objects, we utilize Llama (Touvron et al., 2023) with a tailored prompt (Figure 9). Given the video and audio captions, our prompt instructs the model to generate a single sentence highlighting the common features between the audio and visual content. The prompt emphasizes including only events present in both modalities, while excluding modality-specific details such as overly specific visual features. The model is guided to capture the order and parallel occurrence of events using temporal markers like "and then", "followed by", and "while". This process enhances the consistency between audio and visual descriptions.

**Audio filtering.** We filter out clips containing human vocalizations (e.g., singing, talking), voiceovers, and music using a sound event detection model (Kong et al., 2020b) and the metadata from AudioSet. This step ensures that the remaining audio data largely consists of ambient and context-specific sounds that are more likely to align with the visual content.

After applying these refinement steps, the dataset is reduced to 748 hours of video clips that most likely contain continuous sounds throughout each clip and exhibit high audio-visual correspondence.

> **Role-System:**
> You are a helpful assistant for identifying audio-visual events and generating sentences. Your task is to identify the overlapping or common features between a 10-second audio and the corresponding visual description, and help the user to generate a single sentence of caption that represents this intersection.
> The caption feature is a sentence generated by an audio-caption model: **{enclap_caption}**.
> The label feature is several audio events that happened in the audio: **{audio_label}**.
> Lastly, the user is given several sentences which are the image description of the scene for each second, connected by "and then".
> Please identify all the audio events and visual elements based on all three features and try to conclude in one single sentence to describe this scene with the shared audio-visual events or actions that present sound and sight together.
> Please emphasize time features to present the order of each event, such as "and then", "followed by", "after" for order; "and", "while" etc., for parallel events.
> **Intersection Focus:**
> • Based on the first caption feature, you might need to change or alter any wrong audio event, improve the sentence with more features, such as the weather, the emotion of any people, the description of the car and so on.
> • Keep only the features that are common between the audio and visual descriptions. If an event or element is mentioned in both the audio and the visual description, include it in the final caption.
> • Omit any feature or detail that is present in only one modality. This includes removing overly specific visual details, such as the color, shape, any text or label, name and what people are writing and so on, that do not align with the audio description and vice versa.
> Please ensure that the final caption accurately reflects the common elements of the audio-visual scene, maintaining the order of occurrence, and capturing the shared background, foreground, and context.
> **Role-User:**
> The descriptions of the frames are: **{frame_caption}**

Figure 9: **Prompt for Llama**. We extract common features between the audio and visual caption using Llama, ensuring the resulting caption focuses on events present in both modalities while avoiding overly specific details.

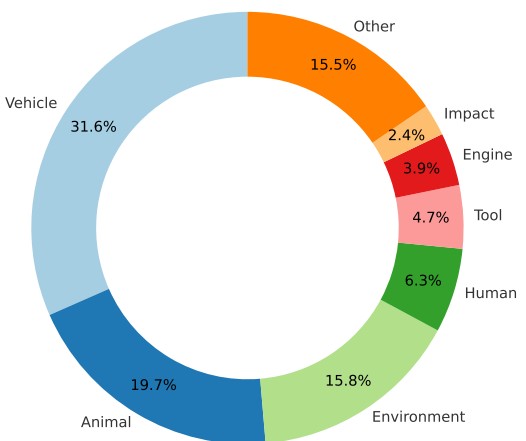

Figure 10: **Categorical distribution of the filtered AudioSet.** We show top 8 categories derived from AudioSet annotations.

### B.2. Dataset Configuration

Figure 10 shows the top-8 categorical distributions, derived from AudioSet (Gemmeke et al., 2017) annotations. We uniformly sample 48 hours across these categories for the test set, with the remaining used for training. Notably, there is no overlap between training and testing videos. As most clips contain multiple sound sources, we randomly select 100 examples from the test set to assess our model's ability to generate object-specific sounds through human evaluation. For 50 of these samples, we manually create object masks by splitting each caption into object snippets and then randomly

selecting one to guide SAM in generating the mask.

## C. Additional Evaluation Details

**ACC.** We use the PANNs model[*] (Kong et al., 2020b) to compute ACC for each audio clip, leveraging annotations from AudioSet. Each audio clip is first processed through the pre-trained PANNs model to obtain logit values for all sound event classes, excluding classes like "Speech" and "Music." We then sample the logits for each clip based on its annotated labels in AudioSet. Since these logits are softmax outputs, they represent the model's confidence in each sound event, allowing us to interpret them as accuracy scores. Finally, we compute the mean of these sampled logits across all clips to determine the overall ACC score.

**FAD, KL, and IS.** We measure FAD, KL, and IS using the AudioLDM-Eval toolbox[†]. The reference and generated audio files are organized into separate folders, and the toolbox is run in paired mode.

**AVC.** We measure AVC using a two-stream network (Arandjelovic & Zisserman, 2017). One stream extracts audio features, while the other extracts visual features. We use OpenL3[‡] (Cramer et al., 2019) to obtain these features and compute the cosine similarity for each image-audio pair.

---

[*] https://github.com/qiuqiangkong/audioset_tagging_cnn
[†] https://github.com/haoheliu/audioldm_eval
[‡] https://github.com/marl/open13

| Scale | ACC (↑) | FAD (↓) | KL (↓) | IS (↑) | AVC (↑) |
|---|---|---|---|---|---|
| $\lambda = 1.0$ | 0.413 | 2.021 | 0.914 | 1.336 | 0.674 |
| $\lambda = 1.5$ | 0.657 | 1.558 | 0.762 | 1.617 | 0.751 |
| $\lambda = 2.0$ | **0.859** | **1.271** | **0.517** | **2.102** | **0.891** |
| $\lambda = 2.5$ | 0.807 | 1.440 | 0.589 | 2.012 | 0.853 |
| $\lambda = 3.0$ | 0.796 | 1.482 | 0.576 | 2.023 | 0.841 |

Table 4: Quantitative results under different CFG scales.

| Threshold | ACC (↑) | FAD (↓) | KL (↓) | IS (↑) | AVC (↑) |
|---|---|---|---|---|---|
| 0.4 | 0.521 | 1.874 | 0.888 | 1.432 | 0.696 |
| 0.5 | 0.743 | 1.536 | 0.691 | 1.625 | 0.774 |
| 0.6 | **0.859** | **1.271** | **0.517** | **2.102** | **0.891** |
| 0.7 | 0.845 | 1.387 | 0.612 | 1.987 | 0.882 |
| 0.8 | 0.812 | 1.501 | 0.664 | 2.005 | 0.879 |

Table 5: Quantitative results under different audio-visual matching scores.

| Method | ACC (↑) | FAD (↓) | KL (↓) | IS (↑) | AVC (↑) |
|---|---|---|---|---|---|
| w/o PE | 0.787 | 1.493 | 0.674 | 1.913 | 0.779 |
| w/ PE (Ours) | **0.859** | **1.271** | **0.517** | **2.102** | **0.891** |

Table 6: Model performance comparison with and without positional encoding.

**Human evaluation.** We conducted a human evaluation to assess the quality and relevance of the generated audio using Amazon Mechanical Turk[§]. The interface for this study is shown in Figure 11. Each participant was presented with an input image and the corresponding generated audio, then rated each sample on a scale from 1 to 5 based on the following criteria: (i) Overall Quality (OVL), assessing the general audio quality; (ii) Relevance to Input Text (RET), measuring the alignment of the audio with the associated text description; (iii) Relevance to Input Image (REI), evaluating how well the audio corresponds to the visual content; and (iv) Relevance to Selected Object (REO), focusing on the alignment of the audio with a specific object in the image.

We randomly selected 100 samples for evaluation, each rated by 50 unique participants to ensure reliability. These samples included both (50%) holistic and (50%) object-specific cases. To control for random responses, we incorporated a set of noise-only samples. Consistently low scores for these control samples confirmed the reliability of participants. Additionally, we ensured that each participant spent at least 90 seconds evaluating each sample to guarantee thoughtful assessment.

To further validate our results, we computed the inter-rater reliability using Cohen's kappa (McHugh, 2012), which indicated a substantial agreement among raters ($\kappa = 0.78$). Furthermore, we conducted a statistical significance test (paired t-test) (Kim, 2015) between our model and baselines for each criterion, confirming that the improvements reported are statistically significant ($p < 0.01$). The final scores presented in the main paper are the mean ratings across all participants.

## D. Additional Results

**Different CFG scales.** We evaluate our model's performance across CFG scales ranging from 1.0 to 3.0. As shown in Table 4, there is a consistent improvement in metrics as $\lambda$ increases from 1.0 to 2.0, reaching peak performance at $\lambda = 2.0$. However, further increasing $\lambda$ beyond 2.0 results

in a gradual decline across most metrics.

**Different thresholds of audio-visual matching.** We test our model's performance across different audio-visual matching thresholds, varying from 0.4 to 0.8 (Figure 7). The same held-out test set is used to assess the metrics, with results presented in Table 5. We empirically find that the model achieves optimal performance at a threshold of 0.6.

**Effect of positional encoding.** We assess the impact of positional encoding (PE) on our model's performance. As shown in Table 6, removing positional encoding leads to a significant degradation across all metrics, highlighting its importance in the model's overall performance.

**Impact of overall scene context.** We examine whether capturing the overall scene context benefits audio generation. To this end, we compare the Captioning & Mix baseline, where each detected object in the image is captioned separately, passed to AudioLDM to generate individual audio clips, and subsequently mixed, against the Captioning baseline (as described in Section 4.1) that leverages the full scene. As shown in Table 7, although Captioning & Mix yields more accurate audio events (ACC), the perceptual metrics (FAD, KL, IS, and AVC) consistently favor the full-scene Captioning method. These results suggest that context awareness is crucial for generating high-quality audio.

**Choice of segmentation module.** We replace SAM with SAM 2 (Ravi et al., 2024), a more sophisticated segmentation method, and evaluate it on the test set. We show in Table 8 that this substitution leads to further gains in generation accuracy and quality, which confirms that more precise segmentation masks benefit our method and aligns well with Theorem 3.1.

**Synchformer-based metric.** Inspired by Synchformer's contrastive pre-training (Iashin et al., 2024), we employ

---

[§]https://www.mturk.com/

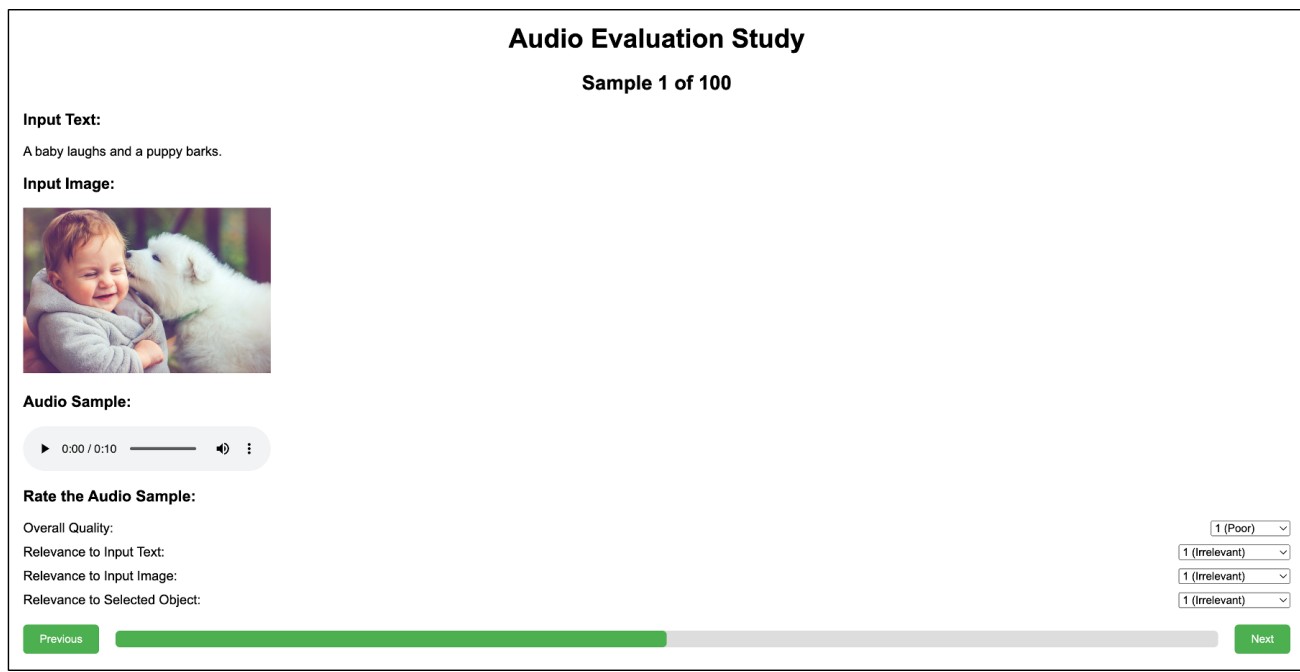

Figure 11: **Human evaluation interface.** We show the interface used for the subjective evaluation of generated audio samples. Participants are presented with input text, an image, and a corresponding audio sample, and are instructed to rate the audio on four criteria. All ratings must be completed before advancing to the next sample.

| Method | ACC (↑) | FAD (↓) | KL (↓) | IS (↑) | AVC (↑) |
|---|---|---|---|---|---|
| Captioning & Mix | **0.643** | 7.634 | 2.511 | 1.443 | 0.645 |
| Captioning | 0.587 | **2.778** | **1.364** | **1.901** | **0.773** |

Table 7: Model performance comparison with and without mixing strategy.

| Method | ACC (↑) | FAD (↓) | KL (↓) | IS (↑) | AVC (↑) |
|---|---|---|---|---|---|
| w/ SAM | 0.859 | 1.271 | 0.517 | 2.102 | 0.891 |
| w/ SAM 2 | **0.881** | **1.153** | **0.472** | **2.295** | **0.936** |

Table 8: Evaluation comparison between SAM and SAM 2 modules.

| Method | IB (↑) |
|---|---|
| Ground Truth | 0.66 |
| Retrieve & Separate (Zhao et al., 2018) | 0.29 |
| AudioLDM 1 (Liu et al., 2023) | 0.24 |
| AudioLDM 2 (Liu et al., 2024) | 0.27 |
| Captioning (Li et al., 2022a) | 0.31 |
| Make-an-Audio (Huang et al., 2023b) | 0.19 |
| Im2Wav (Sheffer & Adi, 2023) | 0.33 |
| SpecVQGAN (Iashin & Rahtu, 2021) | 0.37 |
| Diff-Foley (Luo et al., 2023) | 0.39 |
| Ours | **0.45** |

Table 9: Comparison of ImageBind (IB) scores across different methods.

ImageBind (Girdhar et al., 2023) to measure audio-visual matching on static images. By extracting features from both modalities and computing cosine similarity, we show in Table 9 that our method consistently outperforms baselines on this metric.

# E. Additional Dataset Evaluations

**VGG-Sound dataset.** To further evaluate our method, we evaluate it on the VGG-Sound dataset (Chen et al., 2020), which contains in-the-wild audio-visual data collected from YouTube. Following (Chen et al., 2021a), we use VGG-Sound Sync, a 14-hour subset that contains about 5,000 video clips with better audio-visual synchronization, for testing. To obtain captions aligned with this dataset, we apply the same refinement procedure described in Appendix B.1. We assess our model and all baselines (trained on AudioSet) using the VGG-Sound Sync dataset. As presented in Table 10, our method outperforms all baselines across multiple metrics, particularly in ACC. These results indicate that our model generates more accurate audio that captures the complexity of each scene, while preserving audio quality.

**ImageHear dataset.** We also evaluate our method on the ImageHear dataset (Sheffer & Adi, 2023), an image-to-audio benchmark comprising 100 web-sourced images spanning 30 visual categories (2–8 images per class). Although

| Method | ACC (↑) | FAD (↓) | KL (↓) | IS (↑) | AVC (↑) |
|---|---|---|---|---|---|
| Ground Truth | / | / | / | / | 0.986 |
| Retrieve & Separate (Zhao et al., 2018) | 0.143 | 4.731 | 1.726 | 1.782 | 0.713 |
| AudioLDM 1 (Liu et al., 2023) | 0.256 | 3.876 | 1.634 | 1.901 | 0.634 |
| AudioLDM 2 (Liu et al., 2024) | 0.401 | 3.114 | 1.137 | 1.915 | 0.687 |
| Captioning (Li et al., 2022a) | 0.491 | 2.378 | 1.089 | 2.001 | 0.763 |
| Make-an-Audio (Huang et al., 2023b) | 0.395 | 3.436 | 1.571 | 1.876 | 0.721 |
| Im2Wav (Sheffer & Adi, 2023) | 0.412 | 3.005 | 1.474 | 1.894 | 0.747 |
| SpecVQGAN (Iashin & Rahtu, 2021) | 0.544 | 2.722 | 1.015 | 1.916 | 0.796 |
| Diff-Foley (Luo et al., 2023) | 0.607 | 1.834 | 0.941 | 2.161 | 0.851 |
| **Ours** | **0.761** | **1.112** | **0.675** | **2.342** | **0.898** |

Table 10: Additional quantitative comparison of our method and baselines on the VGG-Sound Sync dataset.

| Method | CS (↑) | ACC (↑) |
|---|---|---|
| Make-an-Audio (Huang et al., 2023b) | 27.44 | 0.77 |
| Im2Wav (Sheffer & Adi, 2023) | 9.53 | 0.49 |
| SpecVQGAN (Iashin & Rahtu, 2021) | 18.98 | 0.49 |
| Diff-Foley (Luo et al., 2023) | 35.12 | 0.86 |
| Ours | 47.37 | 0.88 |

Table 11: Additional comparison of our method and baselines on the ImageHear dataset.

each image contains only a single object, which does not align well with our object-aware setting, our method continues to outperform all baselines in both clip-score (CS) and ACC, as reported in Table 11.

# F. Proof of Theorem 3.1

*Proof.* For notation simplicity, let $u_q \in \Delta^P$ denote the softmax attention weight computed on query $q$ such that $u_{q,l} = \frac{\exp(\langle \mathcal{E}_v(t_q), \mathcal{E}_t(i_{q,l}) \rangle_\Sigma)}{\sum_{k=1}^{P} \exp(\langle \mathcal{E}_v(t_q), \mathcal{E}_t(i_{q,k}) \rangle_\Sigma)}$. We first state the following lemma.

**Lemma F.1.** *Under the same conditions in Theorem 3.1 of the main paper, we have*

$$\mathbb{E}_q[\|u_q - p_q\|_{\ell_1}] \leq \sqrt{2\epsilon_{\text{contrast}}}$$

*Proof.* Notice that

$$
\begin{aligned}
&\epsilon_{\text{contrast}} \\
&= \mathbb{E}_{q,d \sim p_q}\left[ -\log \frac{\exp\left(\langle \mathcal{E}_v(t_q), \mathcal{E}_t(i_{q,d}) \rangle_\Sigma\right)}{\sum_{k=1}^{P} \exp\left(\langle \mathcal{E}_v(t_q), \mathcal{E}_t(i_{q,k}) \rangle_\Sigma\right)} \right] \\
&\quad - \mathbb{E}_{q,d \sim p_q}\left[ -\log p_{q,d} \right] \\
&= \mathbb{E}_{q,d \sim p_q}\left[ \log \frac{p_{q,d}}{u_{q,d}} \right] \\
&= \mathbb{E}_q\left[ D_{\text{KL}}(p_{q,d} \| u_{q,d}) \right]
\end{aligned}
$$

where $D_{\text{KL}}$ denotes the KL distance. By Pinsker's inequal-

ity and Cauchy-Schwarz inequality,

$$
\begin{aligned}
\epsilon_{\text{contrast}} &= \mathbb{E}_q\left[ D_{\text{KL}}(p_{q,d} \| u_{q,d}) \right] \\
&\geq \frac{1}{2} \cdot \mathbb{E}_q\left[ \|p_{q,d} - u_{q,d}\|_{\ell_1}^2 \right] \\
&\geq \frac{1}{2} \cdot \left( \mathbb{E}_q\left[ \|p_{q,d} - u_{q,d}\|_{\ell_1} \right] \right)^2.
\end{aligned}
$$

It follows that

$$\mathbb{E}_q[\|u_q - p_q\|_{\ell_1}] \leq \sqrt{2\epsilon_{\text{contrast}}}.$$

$\square$

Returning to the proof of Theorem 3.1 in the main paper, let $s_q := f(a_q) = f(u_q \boldsymbol{V})$ denote the audio output on query $q$ by the trained model. We decompose $\text{err}_{\text{test}}$ by

$$
\begin{aligned}
&\text{err}_{\text{test}} \\
&= \underbrace{\mathbb{E}_q[v(f^*(p_q \boldsymbol{V}^*), i_q, p_q)] - \mathbb{E}_q[v(f^*(u_q \boldsymbol{V}^*), i_q, p_q)]}_{A} \\
&\quad + \underbrace{\mathbb{E}_q[v(f^*(u_q \boldsymbol{V}^*), i_q, p_q)] - \mathbb{E}_q[v(f^*(a_q), i_q, p_q)]}_{B} \\
&\quad + \underbrace{\mathbb{E}_q[v(f^*(a_q), i_q, p_q)] - \mathbb{E}_q[v(f(a_q), i_q, p_q)]}_{C} \\
&\quad + \underbrace{\mathbb{E}_q[v(f(a_q), i_q, p_q)] - \mathbb{E}_q[v(f(a_q), i_q, m_q)]}_{D} \\
&\quad + \underbrace{\mathbb{E}_q[v(f(a_q), i_q, m_q)] - \mathbb{E}_q[v(f(m_q \boldsymbol{V}), i_q, m_q)]}_{E}.
\end{aligned}
$$

By Lemma F.1 and $\|\boldsymbol{V}^*\|_\infty \leq B_v$, we have

$$
\begin{aligned}
A &\leq \mathbb{E}_q[L_v \cdot L_f \cdot B_v \cdot \|u_q - p_q\|_{\ell_1}] \\
&\leq L_v \cdot L_f \cdot B_v \cdot \sqrt{2\epsilon_{\text{contrast}}}.
\end{aligned}
$$

Since $\|\boldsymbol{V}^* - \boldsymbol{V}\|_\infty \leq \epsilon_{\boldsymbol{V}}$ and $\|u_q\|_1 = 1$, we have

$$
\begin{aligned}
B &= \mathbb{E}_q[v(f^*(u_q \boldsymbol{V}^*), i_q, p_q)] - \mathbb{E}_q[v(f^*(u_q \boldsymbol{V}), i_q, p_q)] \\
&\leq L_v \cdot L_f \cdot \epsilon_{\boldsymbol{V}}.
\end{aligned}
$$

By definition, $C \leq \epsilon_f$. Using the definition $\epsilon_{\text{sam}} = \mathbb{E}_q[\|\boldsymbol{m}_q - p_q\|_{\ell_1}]$, we have

$$D \leq \mathbb{E}_q[L_v \cdot \|\boldsymbol{m}_q - p_q\|_{\ell_1}]$$
$$\leq L_v \cdot \epsilon_{\text{sam}}.$$

and using $\|\boldsymbol{V}\|_\infty \leq B_v$ with Lemma F.1,

$$E \leq \mathbb{E}_q[L_v \cdot L_f \cdot B_v \cdot \|\boldsymbol{m}_q - u_q\|_{\ell_1}]$$
$$\leq L_v \cdot L_f \cdot B_v \cdot (\epsilon_{\text{sam}} + \sqrt{2\epsilon_{\text{contrast}}}).$$

Combining, we have

$$\text{err}_{\text{test}} \leq L_v \cdot L_f \cdot \left(\epsilon_{\boldsymbol{V}} + B_v \cdot \left(\epsilon_{\text{sam}} + 2\sqrt{2\epsilon_{\text{contrast}}}\right)\right)$$
$$+ L_v \cdot \epsilon_{\text{sam}} + \epsilon_f.$$

This completes the proof. $\qquad\square$

