# OpenReview forum: "Sounding that Object: Interactive Object-Aware Image to Audio Generation"
_ICML.cc/2025/Conference — ICML 2025 poster_

### Official Review · Reviewer_wC89 · 2025-03-09

**Overall Recommendation:** 4

**Summary:**

This paper introduces a new object-level video-to-audio generation method that exploits SAM to specify which object in a video should have sound and the AudioLDM architecture to generate the sound. During training, scaled dot-product attention between text embedding and patch-wise image embedding is computed and fed into AudioLDM, while SAM's mask is used during inference

**Claims And Evidence:**

The author claims that exisiting video-to-audio models are designed to generate sound mixtures rather than object-level sounds, which limits creators from manipulating each sound separately. The proposed method performs better than some of the exisiting baselines in terms of audio quality and temporal alignment

**Essential References Not Discussed:**

Video-to-audio generation papers appeared 2024-present are completely missing in Related Works:
- Video-Guided Foley Sound Generation with Multimodal Controls [CVPR'25]
- Taming Multimodal Joint Training for High-Quality Video-to-Audio Synthesis [CVPR'25]
- Temporally Aligned Audio for Video with Autoregression [ICASSP'25]
- STA-V2A: Video-to-Audio Generation with Semantic and Temporal Alignment [ICASSP'25]
- Frieren: Efficient Video-to-Audio Generation Network with Rectified Flow Matching [NeurIPS'24]
- Tell What You Hear From What You See - Video to Audio Generation Through Text [NeurIPS'24]
- FoleyGen: Visually-Guided Audio Generation [MLSP'24]
- V2A-Mapper: A Lightweight Solution for Vision-to-Audio Generation by Connecting Foundation Models [AAAI'24]

Some of the latest text-to-audio generation papers would be nice to include:
- SoundCTM: Unifying Score-based and Consistency Models for Full-band Text-to-Sound Generation [ICLR'25]
- Stable Audio Open [ICASSP'25]
- SpecMaskGIT: Masked Generative Modeling of Audio Spectrograms for Efficient Audio Synthesis and Beyond [ISMIR'24]

Video and/or image-queried sound separation are also relevant:
- OmniSep: Unified Omni-Modality Sound Separation with Query-Mixup [ICLR'25]
- iQuery: Instruments as Queries for Audio-Visual Sound Separation [CVPR'23]
- CLIPSep: Learning Text-queried Sound Separation with Noisy Unlabeled Videos [ICLR'23]

**Experimental Designs Or Analyses:**

The paper mostly follows standard evaluation settings used in the past literature, such as FAD and KL. Audio-visual alignment is evaluated with AVC, which, to my knowledge, has not been used in prior works

**Methods And Evaluation Criteria:**

During training, softmax attention weights between CLIP and CLAP embeddings are computed. In inference, SAM's masks are used instead to condition the audio generative model. Evaluation was done in both objective and subjective ways. AVC was employed to evaluate the temporal alignment between video and audio. In addition to standard metrics such as FAD, IS, a subjective test was conducted to support the evidence of the proposed claim

**Other Comments Or Suggestions:**

- Using SAM2 and/or SAMRAI instead of SAM would be insightful to see if the results improve further by minimizing err_SAM
- Recent relevant works use SyncFormer-based metrics to evaluate audio-visual alignment. I'd like to see the performance in terms of either of these: DeSync proposed in MMAudio, AV-Sync in MultiFoley, or Sync in V-AURA

**Other Strengths And Weaknesses:**

[Strength]
- The proposed idea to use SAM at inference time is very new and would give a straightforward user interface for creators to select objects of interest
- The theory behind the assumption is clear and correct

[Weakness]
- The proposed AVC metric is taken from a model proposed in 2017. I assume SyncFormer-based metrics are more reliable for evaluating audio-visual time alignment

**Questions For Authors:**

Q1. How do you balance the volumes of generated sounds from different objects
Q2. How do you generate off-screen sound in this framework?

**Relation To Broader Scientific Literature:**

Most of the existing literature in this field focuses on mixture generation rather than object-level sound generation, which opens up a new door to the field

**Theoretical Claims:**

Theoritical analysis on the error bound is included in this paper. The derivation of eq. (6), written in Appendix, looks correct. The evaluation results also support this claim

---

> ### Author Rebuttal · Authors · 2025-03-29
>
> We thank the reviewer for their comments and time.
>
> **Clarifying AVC.**
>
> In fact, we employed AVC to measure the semantic (instead of temporal) correspondence between audio and image, since our primary goal is to generate audio aligned with objects from image (not video).
>
> **Why images only?**
>
> Our paper’s main contribution is object-aware, user-driven audio generation based on static images. Focusing on images allows us to cleanly isolate object-to-audio relationships and provide more intuitive user control. While incorporating temporal dynamics (in videos) is a natural extension, it entails additional complexities such as motion tracking and scene changes, which lie beyond our paper’s current scope. However, we agree that expanding to video is an important next step.
>
> **Additional Synchformer-based metric.**
>
> We note that Synchformer specifically targets audio-visual synchronization for video-to-audio generation and does not directly apply to the image-to-audio task. However, inspired by Synchformer’s contrastive pre-training (like DeSync in MMAudio and AV-Sync in MultiFoley), we employ ImageBind [Girdhar et al., CVPR 2023] to measure audio-visual matching on static images. By extracting features from both modalities and computing cosine similarity, we show in the table below that our method consistently outperforms all baselines on this metric.
>
> | Method               | IB (↑) |
> |-----------------------|--------|
> | Ground Truth                    | 0.66   |
> | Retrieve & Separate   | 0.29   |
> | AudioLDM 1           | 0.24   |
> | AudioLDM 2           | 0.27   |
> | Captioning           | 0.31   |
> | Make-an-Audio        | 0.19   |
> | Im2Wav               | 0.33   |
> | SpecVQGAN            | 0.37   |
> | Diff-Foley           | 0.39   |
> | Ours                 | **0.45**   |
>
> **Comparison to SAM 2.**
>
> As suggested, we replace SAM with SAM 2 and evaluate our method on the test set. We show in the table below that this substitution leads to further gains in generation accuracy and quality, which confirms that more precise segmentation masks benefit our method and aligns well with Theorem 3.1 of our paper.
>
> | Method    | ACC (↑) | FAD (↓) | KL (↓) | IS (↑) | AVC (↑) |
> |-----------|---------|---------|--------|--------|---------|
> | w/ SAM    | 0.859   | 1.271   | 0.517  | 2.102  | 0.891   |
> | w/ SAM 2  | **0.881**   | **1.153**   | **0.472**  | **2.295**  | **0.936**   |
>
> **Balancing the volume of different objects.**
>
> We demonstrate in Figure 5 and the demo video of our paper that specifying each object separately tends to assign a similar volume to all sources. However, when multiple objects are selected, our method dynamically accounts for context. For example, if a large car dominates the scene, its siren may overwhelm subtle ambient sounds, creating a more realistic blend instead of flattening everything to equal volume. Moreover, we quantitatively confirm this context-driven behavior in Table 1, 7, and 8 of our paper, where our object-aware method better reflects how certain sources can overpower others or combine to create natural audio events.
>
> **Generating off-screen sound.**
>
> Our current method grounds audio into visible objects via segmentation masks. To incorporate off-screen events, we will add another textual cue (like background music) as extra conditioning for content that lacks a corresponding visual region. This requires no changes to segmentation since off-screen sources are not maskable. While off-screen sound lies outside our paper’s main scope, we agree it is a valuable direction for user-driven, interactive audio generation.
>
> **Missing references.**
>
> Thanks for pointing these out. We will include the suggested references in the revised version. We also clarify that, as noted on the right side of Line 73-95 of our paper, our method differs by enabling object-aware audio generation from static images rather than full video or text-centric approaches. We ground audio in user-selected objects, thus offering fine-grained controllability compared to broader scene-level or textual conditioning alone.

---

> > ### Comment · Reviewer_wC89 · 2025-04-09
> >
> > Thank you for the clarification. The rebuttal comments addressed all my concerns. Therefore, I'd like to raise my score

---

> > > ### Author Response · Authors · 2025-04-09
> > >
> > > Thanks for your support and for raising your score. We appreciate your feedback and will carefully include the additional results in the revised version.

---

### Official Review · Reviewer_XU2w · 2025-03-12

**Overall Recommendation:** 3

**Summary:**

This paper proposed a novel object-aware audio generation model, that supports the interaction with users. This work achieves fine-grained control over which objects, and thus which sounds, are present in the generated audio. Empirical and theoretical validation demonstrating the strong performance of the model and user controllability while maintaining audio quality. The demo videos in the suppl. show some high-quality cases, which further demonstrate the high performance of the proposed framework.

**Claims And Evidence:**

The main claims are two: 1) Interactive object-aware audio generation; and 2) fine-grained control over the objects.

The quantitive and qualitative results can support these claims.

**Essential References Not Discussed:**

[1] Tang, Zineng, et al. "Any-to-any generation via composable diffusion." Advances in Neural Information Processing Systems 36 (2023): 16083-16099.

[2] Tang, Zineng, et al. "Codi-2: In-context interleaved and interactive any-to-any generation." Proceedings of the IEEE/CVF Conference on Computer Vision and Pattern Recognition. 2024.

**Experimental Designs Or Analyses:**

Refer to the "Methods And Evaluation Criteria".

**Methods And Evaluation Criteria:**

**Method:**

* The proposed framework demonstrates strong performance in generating audio based on a selected object. However, an important question arises: Can the model generate audio from more than two selected objects? Additionally, can the model simulate interactions among these selected objects? This capability needs further exploration and analysis.
* Regarding the Multi-Layer Perceptron (MLP) designed within the framework, the author should provide a more comprehensive analysis of its effectiveness. Specifically, if the MLP is used merely to map the latent features of the Image Encoder to the latent space of the diffusion model, why not use a simpler linear layer instead? A comparative evaluation could clarify the necessity of using an MLP in this context.

**Evaluation:**

* The paper omits several important baseline models, such as CoDi [1] and CoDi-2 [2], which are relevant for a more robust comparison. Including these models would strengthen the evaluation and provide a clearer benchmark for performance.

* For the human evaluation, the key purpose of such an evaluation is to validate user satisfaction with interacting with the model. However, the current experiment does not effectively fulfill this goal. To better validate the interaction capabilities, I recommend conducting more user studies in the rebuttal, particularly those aimed at evaluating interaction satisfaction.

[1] Tang, Zineng, et al. "Any-to-any generation via composable diffusion." Advances in Neural Information Processing Systems 36 (2023): 16083-16099.

[2] Tang, Zineng, et al. "Codi-2: In-context interleaved and interactive any-to-any generation." Proceedings of the IEEE/CVF Conference on Computer Vision and Pattern Recognition. 2024.

**Other Comments Or Suggestions:**

If my problem is solved well, I will raise my score.

**Other Strengths And Weaknesses:**

Please refer to "Methods And Evaluation Criteria".

**Questions For Authors:**

Please refer to "Methods And Evaluation Criteria".

**Relation To Broader Scientific Literature:**

None

**Theoretical Claims:**

There is a clear proof of the proposed technology.

---

> ### Author Rebuttal · Authors · 2025-03-29
>
> We thank the reviewer for their comments and time.
>
> **Generating audio from multiple objects.**
>
> In fact, we showed in Figure 5 and the demo video of our paper that our method accepts multi-object masks (including more than two objects) to generate an audio mixture that reflects each selected object. We also evaluated a multi-object subset in human perceptual study (the last four columns of Table 1 in our paper), further confirming that our model can handle multiple sources in a single scene.
>
> **Interactions among multiple objects.**
>
> We illustrated in Figure 6 and the demo video of our paper that a single image showing a stick in water or grass can generate splashing or rustling by leveraging full-image segmentation masks. These results indicate our method’s ability to handle basic multi-object interactions from static images. However, we also note that more intricate physical interactions often rely on temporal cues, which are generally studied in video and lie beyond our current single-image scope.
>
> **Linear vs. MLP.**
>
> Our method used a lightweight MLP to capture nonlinear cross-modal interactions more effectively than a simpler linear layer. As shown in the first table below, retraining with a linear projection yields lower performance. Importantly, our MLP added roughly 1% more parameters to the AudioLDM backbone (see the second table below), resulting in a minimal additional computation cost.
>
> | Method     | ACC (↑) | FAD (↓) | KL (↓) | IS (↑) | AVC (↑) |
> |------------|---------|---------|--------|--------|---------|
> | Linear     | 0.811   | 1.432   | 0.605  | 1.974  | 0.876   |
> | MLP (Ours) | **0.859**   | **1.271**   | **0.517**  | **2.102**  | **0.891**   |
>
> | Method      | #Param | MACs  | FLOPs |
> |-------------|--------|-------|-------|
> | AudioLDM    | 317M   | 317M  | 587M  |
> | Linear      | 0.26M  | 0.26M | 0.5M  |
> | MLP (Ours)  | 3.15M  | 3.15M | 6.29M |
>
> **Comparison to CoDi.**
>
> As suggested, we evaluate CoDi (we do not compare CoDi 2 since it is not open-source) on our test set in the table below. We find that our model still holds better performance across all metrics. Unlike CoDi’s global any-to-any generation, our model explicitly allows users to select specific objects within an image for fine-grained control in audio generation.
>
> | Method | ACC (↑) | FAD (↓) | KL (↓) | IS (↑) | AVC (↑) |
> |--------|---------|---------|--------|--------|---------|
> | CoDi   | 0.672   | 1.954   | 0.856  | 1.936  | 0.833   |
> | Ours   | **0.859**   | **1.271**   | **0.517**  | **2.102**  | **0.891**   |
>
> **Interaction satisfaction evaluation.**
>
> As suggested, we conduct a human study focusing on user-driven audio generation, comparing our method to text-based baselines (we exclude image- and video-based baselines, as they do not allow user prompting). We ask 5 experienced participants to generate "baby laughs and puppy barks" from a single image (Figure 2 of our paper), and we measure the average time taken (in minutes), the number of attempts required, and a 5-point subjective satisfaction score (with 95% confidence intervals). As shown in the table below, text-based baselines often miss one of the sounds and require multiple prompt adjustments, leading to higher time and lower satisfaction. Our method, by contrast, consistently requires fewer attempts, takes less time, and achieves higher satisfaction—even for participants already familiar with prompting. We will include these findings in the revised version.
>
> | Method     | Time (↓) | Attempts (↓) | Satisfaction (↑) |
> |------------|----------|--------------|-------------------|
> | AudioLDM 1 | 7.34     | 3.20         | 2.00 ± 0.88       |
> | AudioLDM 2 | 5.10     | 2.40         | 2.80 ± 1.04       |
> | Ours       | **2.67**     | **1.60**         | **3.60 ± 0.68**       |
>
> **Missing references.**
>
> Thanks for pointing these out. We will include CoDi and CoDi 2 in the revised version. We also clarify that, as noted on the right side of Line 73-95 of our paper, our method differs by enabling object-aware audio generation in static images rather than full video or text-centric approaches. We ground audio in user-selected objects, thus offering fine-grained controllability compared to CoDi’s global any-to-any generation.

---

> > ### Comment · Reviewer_XU2w · 2025-04-08
> >
> > Thanks for the rebuttal. The additional results look good to me. Since my original score is weak accept, my score will remain the same. I hope the authors can revise the paper as I suggested in the final version if accepted.

---

> > > ### Author Response · Authors · 2025-04-08
> > >
> > > We appreciate your feedback and are glad that you found our additional results helpful. We will carefully include your comments in a revision. If our rebuttal strengthens your confidence in our paper, we would greatly appreciate your consideration to increase the score.

---

### Official Review · Reviewer_kpPn · 2025-03-13

**Overall Recommendation:** 3

**Summary:**

This paper introduces an object-aware image-to-audio generation framework built on top of pretrained AudioLDM. Given user-provided segmentation mask, the I2A generation method can generate the object-aligned sound. Experiments on AudioSet and VGGSound Sync datasets show the proposed method outperforms selected vision-to-audio generation algorithms in both objective and subjective evaluation. They also showcase the model's generation control by interactively editing the mask of visual objects in a soundscape scene.

**Claims And Evidence:**

The claim of object-aware image-to-audio generation is supported with experiments on AudioSet and VGGSound Sync.

However, I'm curious about how naive method could solve the problem. For example, what if we use a captioning model to describe the user-selected region and then put the description to Text-to-Audio generation model such as AudioLDM?

I understand there are subtle visual details that cannot be described by captioning model. Then a comparison to similar object-level image-to-audio generation approaches might be necessary. The compared methods are all scene-level V2A models which do not address object-level sound generation as intended in the proposed task. Although they are not intended to solve interactive object-level I2A, it seems SSV2A [1] also has the ability to generate object-level sound when specifying the visual region.

[1] Guo et al., Gotta Hear Them All: Sound Source Aware Vision to Audio Generation, 2024

**Essential References Not Discussed:**

The paper introduces object-aware image-to-audio generation but misses the mention of a highly related work SSV2A [1].

[1] Guo et al., Gotta Hear Them All: Sound Source Aware Vision to Audio Generation, 2024

Also citations to some seminal works in V2A are missing:

[3] V2A-Mapper: A Lightweight Solution for Vision-to-Audio Generation by Connecting Foundation Models, AAAI 2024

[4] Seeing and Hearing: Open-domain Visual-Audio Generation with Diffusion Latent Aligners, CVPR 2024

**Experimental Designs Or Analyses:**

I'm curious about how naive method could solve the problem. For example, what if we use a captioning model to describe the user-selected region and then put the description to Text-to-Audio generation model such as AudioLDM?

I understand there are subtle visual details that cannot be described by captioning model. Then a comparison to similar object-level image-to-audio generation approaches might be necessary. The compared methods are all scene-level V2A models which do not address object-level sound generation as intended in the proposed task. Although they are not intended to solve interactive object-level I2A, it seems SSV2A [1] also has the ability to generate object-level sound when specifying the visual region.

[1] Guo et al., Gotta Hear Them All: Sound Source Aware Vision to Audio Generation, 2024

**Methods And Evaluation Criteria:**

The evaluation metrics make sense in evaluating the alignment between the object and generated sound.

However, since the authors propose to solve the image-to-audio generation task, experiments instead were performed on video datasets where image frame is selected. How about the performance on image dataset ImageHear[2]?

[2] Sheffer and Adi, I Hear Your True Colors: Image Guided Audio Generation, ICASSP 2023

**Other Comments Or Suggestions:**

No.

**Other Strengths And Weaknesses:**

The paper is well presented and easy to follow.
I appreciate the demo videos which are very helpful in understanding the quality of the generated sound.
The proposed method seems simple and effective in grounding object representation in audio diffusion model.

However my main concern is about the experimental design/analysis where a critical comparison to a naive method and a highly-related method is missing. The work could be more convincing if including this.

**Questions For Authors:**

I'm curious about how naive method could solve the problem. For example, what if we use a captioning model to describe the user-selected region and then put the description to Text-to-Audio generation model such as AudioLDM?

I understand there are subtle visual details that cannot be described by captioning model. Then a comparison to similar object-level image-to-audio generation approaches might be necessary. The compared methods are all scene-level V2A models which do not address object-level sound generation as intended in the proposed task. Although they are not intended to solve interactive object-level I2A, it seems SSV2A [1] also has the ability to generate object-level sound when specifying the visual region.

[1] Guo et al., Gotta Hear Them All: Sound Source Aware Vision to Audio Generation, 2024

**Relation To Broader Scientific Literature:**

This paper proposes to solve object-aware image-to-audio generation where user can specify random regions by providing segmentation mask. I'm concerned about the necessity of the solution. Instead of introducing any training, what's the performance of just using existing captioning model to describe the masked region and then call a T2A model? According to the demo videos, there seems no correlation in identity between the sound generated by single object region and the larger region including the object. Then it seems text prompt could already suffice.

**Theoretical Claims:**

The theoretical analysis is neatly presented and looks good to me.

---

> ### Author Rebuttal · Authors · 2025-03-29
>
> We thank the reviewer for their comments and time.
>
> **Clarifying caption-based methods.**
>
> In fact, we have evaluated two caption-based variants in Table 1 & 7 of our paper. In Captioning, we generated a single caption from the entire image and fed it to a pre-trained AudioLDM 2. In Captioning & Mix, we generated separate captions for each detected object, synthesized individual audio clips with AudioLDM 2, and mixed them. As expected, both methods perform worse than ours, but interestingly Captioning outperforms Captioning & Mix, suggesting that caption-based methods do not fully address context or proportional mixing—an observation also noted on the right side of Line 45-48 in our paper.
>
> **Region-based audio correlation.**
>
> We demonstrate in Figure 5 and the demo video of our paper that naive region-based layering tends to assign similar loudness to all sources, leading to flat, unrealistic soundscapes. In contrast, our method dynamically accounts for context.  For example, if a large car dominates the scene, its siren may overwhelm subtle ambient sounds, creating a more realistic blend instead of flattening everything to equal volume. Likewise, Figure  6 of our paper shows that our method captures interactions—like a stick splashing water—instead of generating only generic water flowing sounds. Moreover, we quantitatively confirm this context-driven behavior on AudioSet and VGGSound Sync in Table 1 & 8 of our paper, where our object-aware method better reflects how certain sources can overpower others or combine to create natural audio events.
>
> **Clarifying SSV2A.**
>
> SSV2A is an unpublished concurrent work and the code has not been fully released before the submission deadline, so we did not compare with it, per ICML policy. In fact, our method is quite different from SSV2A: (1) our training is self-supervised, while they require bounding boxes from external object detectors such as YOLOv8; (2) we allow fine-grained user control on objects, while their manifold pipeline lacks this capability. As suggested, we now evaluate SSV2A on our test set, showing that we generate more accurate and comparable high-quality audio:
>
> | Method | ACC (↑)| FAD (↓) | KL (↓) | IS (↑) |AVC (↑)|
> |-|-|-|-|-|-|
> |SSV2A|0.806|**1.265**|0.525|2.100|**0.893**|
> |Ours|**0.859**|1.271|**0.517**|**2.102**|0.891|
>
> **Additional dataset evaluation.**
>
> The ImageHear dataset contains only a single object per image, which does not align well with our object-aware setting so we did not evaluate on it. However, as suggested, we now compare our method to both image- and video-based baselines using metrics from ImageHear. As shown in the table below, our method continues to outperform them:
>
> | Method         | CS (↑) | ACC (↑)  |
> |----------------|--------|----------|
> | Make-an-Audio  | 27.44  | 77.31%   |
> | Im2Wav         | 9.53   | 49.14%   |
> | SpecVQGAN      | 18.98  | 48.76%   |
> | Diff-Foley     | 35.12  | 86.45%   |
> | Ours           | **47.37**  | **88.32%**   |
>
> **Missing references.**
>
> Thanks for pointing these out. We will include these references in the revised version. We also clarify that, as noted on the right side of Line 73-95 of our paper, our method centers on user-selected object-aware audio generation in static images, allowing finer control than the concurrent work SSV2A’s bounding-box method (as discussed above) and the broader scene-level approaches like V2A-Mapper or Seeing & Hearing.

---

### Official Review · Reviewer_MZUs · 2025-03-14

**Overall Recommendation:** 3

**Summary:**

This paper proposes an image-to-audio generation method with interactive object-aware design.
It mainly concentrates on decoupling separate events in visual scenes, while processing the overall scene context.
To train the visual object grounding model, the attention module is designed and substituted with a user-selected mask at inference.
The authors also provide a theoretical analysis for such a substitution.
Experiments provide various ablation analyses and comparisons with baselines.

**Claims And Evidence:**

The problem claimed by the authors is valid, and they propose an interesting approach to incorporating user interaction into the visual-to-sound generation task.

**Essential References Not Discussed:**

Recent visual-to-sound generation methods have been missed in this paper.
The authors discussed previous approaches that were published until 2023.
This is my primary reason for rejection, as demonstrating superiority through performance comparisons with recent methods is crucial.

[1] Y. Xing et al., "Seeing and Hearing: Open-domain Visual-Audio Generation with Diffusion Latent Aligners".

[2] Y. Jeong et al., "Read, Watch and Scream! Sound Generation from Text and Video".

[3] Z. Xie et al., "Sonicvisionlm: Playing sound with vision language models".

[4] Y. Zhang et al., "Foleycrafter: Bring silent videos to life with lifelike and synchronized sounds".

**Experimental Designs Or Analyses:**

The ablation study and analysis provided by the authors, particularly Table 2, serve as strong evidence to support their claims.
However, the primary baselines used for comparison are methods published up until 2023 (except for AudioLDM2, but it is a text-to-audio method), and there is a lack of performance comparison with more recent approaches.
Additionally, the evaluation is limited to a constrained set of datasets.
The authors follow multiple steps to refine the datasets, similar to Sound-VECaps, but evaluating the model on such a highly curated dataset may be insufficient to demonstrate its generalization capability.

**Methods And Evaluation Criteria:**

One concern is that this cross-modal attention approach is no longer novel and relies on a well-trained, high-performance masked model during testing, which could lead to significant computational overhead. Another concern is that a similar method [1] already exists, making it necessary to clarify the methodological and experimental differences between this work and previous studies.
The evaluation metrics are appropriate. However, it is unclear how existing video-based approaches have been modified in this work.

[1] W. Guo et al., "Gotta Hear Them All: Sound Source-Aware Vision to Audio Generation"

**Other Comments Or Suggestions:**

n/a

**Other Strengths And Weaknesses:**

n/a

**Questions For Authors:**

1. Recent approaches (see `Essential References Not Discussed') utilize not only video but also text as input.
Since this method also processes textual information, a comparison with such approaches is necessary.

2. Moreover, recent methods that utilize AudioLDM typically freeze the diffusion model during use.
A comparison of such approaches such as [1] and [2] with its frozen diffusion (Table 2(i)) is also necessary.
I am also curious about why the FAD score of the frozen model in Table 2 differs significantly from the original model.

3. If I was not missing this part, I recommend explaining how the video-based model was processed.


[1] W. Guo et al., "Gotta Hear Them All: Sound Source-Aware Vision to Audio Generation".
[2] Y. Jeong et al., "Read, Watch and Scream! Sound Generation from Text and Video".

**Relation To Broader Scientific Literature:**

The paper addresses the training of an object-aware (grounding) module for interactive audio generation. However, the proposed method is limited to images rather than videos. To enhance its applicability, an extended solution for video processing is needed.

**Theoretical Claims:**

I don't have tackling points for theoretical analysis (sec.3.3).

---

> ### Author Rebuttal · Authors · 2025-03-29
>
> We thank the reviewer for their comments and time.
>
> **Reliance on SAM.**
>
> First, our method does not fundamentally rely on SAM for its performance, but instead benefits from it functionally for enhancing user interactivity.  As shown in Table 2(iv), Theorem 3.1, and Figure 4 of our paper, comparable results can be obtained when SAM is replaced with text-image cross-modal attention at test time. Second, we respectfully disagree with the notion that leveraging strong pre-trained models weakens the contribution. Our use of SAM follows a common and productive paradigm in the community to build upon powerful existing components—examples include LLaVA (LLaMA + visual encoder), ControlNet (Stable Diffusion + extra conditioning), and Prompt-to-Prompt (Stable Diffusion + cross-attention control).
>
> **Limited novelty.**
>
> Our core novelty does not lie in proposing a new network architecture or method purely for state-of-the-art performance. Instead, our contributions are: (1) identifying and posing a surprising self-supervised learning signal—distinct objects and their associated sound—for audio generation, which is in the tradition of ICML papers like SimCLR [Chen et al., ICML 2020] that opened up promising research directions; (2) fundamentally overcoming the challenge of forgetting or binding sound events—by introducing multi-modal dot-product attention for audio generation, grounded in theory; (3) enabling a new interactive setup, where users can control which objects produce sound via simple mouse clicks.
>
> **Clarifying SSV2A.**
>
> SSV2A is an unpublished concurrent work and the code has not been fully released before the submission deadline, so we did not compare with it, per ICML policy. In fact, our method is quite different from SSV2A: (1) our training is self-supervised, while they require bounding boxes from external object detectors such as YOLOv8; (2) we allow fine-grained user control on objects, while their manifold pipeline lacks this capability. As suggested, we now evaluate SSV2A on our test set (the table below), showing that we generate more accurate and comparable high-quality audio.
>
> | Method | ACC (↑) | FAD (↓) | KL (↓) |IS (↑) |AVC (↑)|
> |-|-|-|-|-|-|
> |SSV2A|0.806|**1.265**|0.525|2.100|**0.893**|
> |Ours|**0.859**|1.271|**0.517**|**2.102**|0.891|
>
> **Comparison to additional methods.**
>
> As suggested, we compare several video (or video + text) based methods (Seeing & Hearing, ReWaS, and FoleyCrafter), noting that ReWaS is concurrent, FoleyCrafter is unpublished, and SonicVisionLM has no public codebase so we do not compare to it. Since these works do not provide training codes, we instead run their public inference scripts on our test set. The table below shows that our method outperforms these baselines, largely because it introduces object-level specificity.
>
> | Method            | ACC (↑) | FAD (↓) | KL (↓)  | IS (↑) | AVC (↑) |
> |-|-|-|-|-|-|
> | Seeing & Hearing|0.668|1.923   | 0.794   | 1.954  | 0.722   |
> | ReWaS|0.694|1.552|1.134| 1.938  | 0.704|
> | FoleyCrafter      |0.732|1.760| 0.665   | 2.007  | 0.811|
> | Ours              |**0.859**|**1.271**|**0.517**|**2.102**|**0.891**|
>
> **Constrained dataset evaluation.**
>
> We have provided quantitative evaluations on the commonly used AudioSet and VGGSound Sync (Tables 1 & 8 of our paper), which feature in-the-wild videos, and we include out-of-distribution examples (Section 4.4 of our paper) plus a demo video (Appendix A), using inputs from the internet or different datasets (Places & the Greatest Hits). These studies, conducted across diverse datasets at scale, demonstrate our model’s generalization capability.
>
> **Extending to video.**
>
> Thanks for pointing out this important future direction and broader impact. Our paper’s main contribution is object-aware, user-driven audio generation based on static images. Images allow us to cleanly isolate object-to-audio relationships and provide more intuitive user control, while temporal dynamics (in videos) entail additional complexities such as motion tracking and scene changes, which lie beyond our paper’s current scope.
>
> **Comparison to frozen-diffusion.**
>
> Neither SSV2A nor ReWaS reports experiments contrasting frozen vs. fine-tuned diffusion, so they do not quantify how full adaptation might improve cross-modal alignment. The ablation in Table 2(i) of our paper reveals that freezing simplifies training but hinders fine-grained alignment (especially at the object level), leading to higher FAD. By fully fine-tuning, we reduced FAD and improved ACC because the model can capture richer object-specific cues.
>
> **Clarifying video-based baselines.**
>
> As described on the right side of Line 288-289 of our paper, we randomly sampled a single frame from each video clip as input and fine-tuned them on our dataset.
>
> **Missing references.**
>
> Thanks for pointing these out. We will add comparisons (see above) and discussions (the right side of Line 73–95 of our paper) on the suggested papers in the revised version.

---

> > ### Comment · Reviewer_MZUs · 2025-04-05
> >
> > Most of my concerns have been solved, so I am raising the final rating. Also, I recommend including new results and discussion in the revised version.

---

> > > ### Author Response · Authors · 2025-04-05
> > >
> > > Thanks for your support and for raising your score. We appreciate your feedback and will carefully incorporate your suggestions in a revision.

---

### Decision · Program_Chairs · 2025-05-01

**Decision:**

Accept (poster)

**Comment:**

All reviewers recommend accepting the paper (1 accept and 3 weak accepts).

The rebuttal did a good job of addressing the concerns that the reviewers had.

The AC thus follows this unanimous recommendation by the reviewers and recommends accepting the paper.